# Impacts of multisectoral cash plus programs after four years in an urban informal settlement: Adolescent Girls Initiative-Kenya (AGI-K) randomized trial

Beth Kangwana[1]*, Karen Austrian[1], Erica Soler-Hampejsek[2], Nicole Maddox[3], Rachel J. Sapire[4], Yohannes Dibaba Wado[5], Benta Abuya[5], Eva Muluve[1], Faith Mbushi[1], Joy Koech[6], John A. Maluccio[7]

1 Poverty, Gender and Youth Program, Population Council – Kenya, Nairobi, Kenya, 2 Independent Consultant, Barcelona, Spain, 3 Department of Social and Political Sciences, Bocconi University, Milan, Italy, 4 Department of Population and Family Health, Mailman School of Public Health, Columbia University, New York, New York, United States of America, 5 African Population and Health Research Center, Nairobi, Kenya, 6 Population Services International, Kenya, Nairobi, Kenya, 7 Department of Economics, Middlebury College, Middlebury, Vermont, United States of America

* bkangwana@popcouncil.org

**Data Availability Statement:** All data files are currently available from the the HARVARD

## Abstract

### Background

The vast majority of adolescent births occur in low- and middle-income countries and are associated with negative outcomes for both the mother and her child. A multitude of risk factors may explain why few programs have been successful in delaying childbearing and suggest that multisectoral interventions may be necessary. This study examines the longer-term impact of a two-year (2015–17) multisectoral program on early sexual debut and fertility in an urban informal settlement in Kenya.

### Methods

The study used a randomized trial design, longitudinally following 2,075 girls 11–14 years old in 2015 until 2019. The interventions included community dialogues on unequal gender norms and their consequences (violence prevention), a conditional cash transfer (education), health and life skills training (health), and financial literacy training and savings activities (wealth). Girls were randomized to one of four study arms: 1) violence prevention only (V-only); 2) V-only and education (VE); 3) VE and health (VEH); or 4) all four interventions (VEHW). We used ANCOVA to estimate intent-to-treat (ITT) impacts of each study arm and of pooled study arms VE, VEH, and VEHW relative to the V-only arm, on primary outcomes of fertility and herpes simplex virus-2 (HSV-2) infection, and secondary outcomes of education, health knowledge, and wealth creation. Post-hoc analysis was carried out on older girls who were 13–14-years-old at baseline. In 2018, in the VEHW arm, in-depth qualitative evaluation were carried out with adolescent girls, their parents, school staff, mentors, community

Dataverse: https://dataverse.harvard.edu/dataset.xhtml?persistentId=doi:10.7910/DVN/94U224.

**Funding:** Awardee initials: KA Grant Number: PO6171 Funder: Foreign Commonwealth and Development Office URL:https://www.gov.uk/government/organisations/foreign-commonwealth-development-office The funder approved the trial design and provided support in the form of salaries for authors [BK, KA, ESH, NM, YDW, BA, EM, FM, JK and JAM] but did not have any additional role in study design, data collection and analysis, preparation of the manuscript or decision to publish. The funders had no role in study design, data collection and analysis, decision to publish, or preparation of the manuscript.

**Competing interests:** Erica Soler-Hampejsek is a self-employed independent research consultant and not associated with any commercial company. Therefore, we declare there is no commercial affiliation.

conversation facilitators, and community gatekeepers. The trial is registered at ISRCTN: ISRCTN77455458.

## Results

At endline in the V-only study arm, 21.0 percent of girls reported having had sex, 7.7 percent having ever been pregnant and 6.6 percent having ever given birth, with higher rates for the older subsample at 32.5 percent, 11.8 percent, and 10.1 percent, respectively. In the full sample, ever having given birth was reduced by 2.3 percentage points (pp) in the VE and VEHW study arms, significant at 10 percent. For the older subsample there were larger and significant reductions in the percent ever having had sex (8.2 pp), HSV-2 prevalence (7.5 pp) and HSV-2 incidence (5.6 pp) in the VE arm. Two years after the end of the interventions, girls continued to have increased schooling, sexual and reproductive health knowledge, and improved financial savings behaviors. Qualitatively, respondents reported that girls were likely to have sex as a result of child sexual exploitation, peer pressure or influence from the media, as well as for sexual adventure and as a mark of maturity.

## Conclusion

This study demonstrates that multisectoral cash plus interventions targeting the community and household level, combined with interventions in the education, health, and wealth-creation sectors that directly target individual girls in early adolescence, generate protective factors against early pregnancy during adolescence. Such interventions, therefore, potentially have beneficial impacts on the longer-term health and economic outcomes of girls residing in impoverished settings.

## Clinical trial registration

ISRCTN registry: ISRCTN77455458; https://doi.org/10.1186/ISRCTN77455458.

## Background

Adolescence is a period of rapid physical, cognitive, social, emotional and sexual development. These developmental changes, in association with negative external factors including lack of economic security, unequal gender norms, pressure from peers to engage in sexual activity, pressure from families to achieve economic security through early marriage, and not living with one's parents are likely to increase the risk of early pregnancy [1–5]. Girls residing in impoverished settings have greater exposure to these negative external factors and are therefore significantly more likely to engage in unprotected sex at early ages and become pregnant [6]. Globally, 11 percent of all births are to adolescent girls 15–19 years old, with the vast majority (95 percent) of the births occurring in low- and middle-income countries [7].

In Kenya, one in every five girls between 15–19 years is either pregnant or already has a child [8]. Socioeconomically, adolescent childbearing is likely to result in reduced schooling and human capital investment which in turn is likely to lead to reduced job tenure, earnings, and economic empowerment [9]. From a health perspective, complications during pregnancy and childbirth are leading causes of death for females ages 15–24 years [7].

Early sexual initiation is also a known risk factor for herpes simplex virus type 2 (HSV-2) infection in females which has been shown to increase susceptibility to HIV infection two- to threefold and transmission of HIV infection up to fivefold [10–13]. Other risk factors for HSV-2 infection include having multiple sexual partners or a history of other sexually transmitted infections (STIs) [12, 13]. Globally, HSV-2 prevalence in 15–19-year-old adolescents is estimated at 7 percent but is as high as 27 percent in rural regions of Western Kenya [14, 15]. Although in rare instances it can be transmitted to neonates during delivery (typically resulting in severe disability or neonatal death) [16, 17]), HSV-2 is almost exclusively sexually transmitted. Because infection leads to the lifelong production of HSV-2 antibodies, their presence is used as a biological marker of prior sexual behavior [18].

Education has been shown to be a critical protective factor in delaying adolescent sexual debut and preventing unintended pregnancy. In Kenya, for example, completing secondary education or higher was found to have reduced the odds of adolescent pregnancy by 67 percent [19]. Studies consistently link education to improved reproductive health outcomes including delayed age at first birth, and demonstrate that school enrollment may be more effective in reducing adolescent childbearing than other reproductive health education interventions [20]. Dropping out of school, however, can be both a result and a cause of early pregnancy, making the pathways through which education and adolescent pregnancy are linked complex. As in many contexts, in Nairobi's informal settlements there is evidence of a bidirectional association between education and pregnancy, with both higher rates of school dropout among pregnant adolescents, as well as lower likelihood that those enrolled in school will get pregnant [21, 22].

Low socioeconomic status is a risk factor for early pregnancy because adolescents with limited access to economic resources are less able to afford basic care and family planning and are more vulnerable to experiencing child sexual exploitation [5, 23, 24], including having sex in exchange for money or gifts. Cash transfers are a method of promoting economic empowerment and have been shown to increase women's decision-making power and choices regarding marriage, fertility, and engaging in risky sexual activity [25, 26]. Cash transfers have also been demonstrated to improve school enrollment among adolescent girls which, as described above, has shown to be a protective factor against early pregnancy [25, 26]. There is limited and more mixed evidence, however, on the effectiveness of cash transfers alone versus "cash plus" approaches that combine transfers with additional supportive programming, with some studies indicating that cash transfers alone are insufficient to reduce adolescent pregnancy [26, 27]. Both conditional and unconditional cash transfers have been implemented in a wide array of health interventions with the addition of conditionality not always resulting in better outcomes [22, 25].

Some multisectoral interventions that address multiple areas, such as socioeconomic status, education, and health, and that are therefore able to target overlapping vulnerabilities have been shown to be effective in reducing early pregnancy [28]. Relatedly, in South Africa, combining a monthly grant with other structural and behavioral interventions targeting the caregiver and adolescent led to reductions in HIV incidence [29]. In Tanzania, an intervention consisting of cash plus behavioral intervention addressing gender-equitable attitudes improved attitudes among male participants [30]. In addition, there is evidence that early adolescence is a critical window during which to intervene and prevent the potential negative consequences of pregnancies before they occur. Intervening during this critical period, before negative outcomes crystallize, can improve the well-being of the target population as well as of their offspring, interrupting the transmission of poverty [31–33].

The Adolescent Girls Initiative-Kenya (AGI-K) was a randomized trial designed to test the short-term (after two years) and longer-term (after four years) effects of two-year,

multisectoral and multilevel "cash plus" programs for young adolescent girls 11–14 years old in two different marginalized areas of Kenya where they face many of the above challenges: 1) Kibera, an urban informal settlement in Nairobi and 2) rural Wajir County on the northeastern border with Somalia. In this study we examine the effects of AGI-K in Kibera after four years (i.e., two years after the end of the program when the girls were 15–18 years old), on the primary outcome of delayed childbearing, as well as a range of secondary outcomes. Short-term results and results after four years for Wajir are reported elsewhere [34, 35].

## Methods

### Intervention context

Kibera is the largest urban informal settlement in Kenya and is characterized by high population density (more than 20,000 people per square kilometer) alongside substantial residential mobility, and has low-quality housing, high crime rates, minimal government services, and multiple religious and ethnic groups, though large pluralities identify as Luhya (38 percent) and Luo (28 percent) [36]. The characteristics and deprivations of Kibera are similar to other urban settings in Kenya [37] as well as to other urban areas in Africa where more than 50 million people live [38]. Forty-three percent of girls 15–19 years old living in urban informal settlements are not in school, mainly because of their inability to pay school fees. In 2006, 60 percent of adolescent girls 10–19 years old in Kibera felt there was a lot of crime in their neighborhood and feared they would be sexually assaulted [39]. Although the median age of first marriage in Nairobi informal settlements is 22 for 25–39 year-olds, one-quarter of 20–24 year-olds initiated sexual activity by the time they were 16 years old [8, 39]. Childbearing among 15–17-year-olds in informal settlements is much higher than in the rest of Nairobi, as is the portion reporting that their pregnancy was unintended, which was approximately one-half in informal settlements compared to just over one-third in the rest of Nairobi [36]. Fig 1 presents the enrollment of girls into the study, described in more detail below.

### Theory of change

Fig 2 presents the theory of change underpinning the multisectoral, multilevel cash plus AGI-K programs [40]. This theory was originally based on a combination of an asset-building theory of change—that posits that girls need a combination of education, social, health, and economic assets to make a safe, healthy, and productive transition from childhood into young adulthood [41, 42]—as well as an ecological framework for adolescent health that takes into account the various levels in society that shape those outcomes [43]. The figure outlines how interventions in the violence prevention and education sectors targeting the community and the household are combined with interventions in the education, health, and wealth-creation sectors targeting the individual girl. The interventions are designed to work in concert to empower the girls, improving their "ability to formulate strategic choices, and to control resources and decisions that affect important life outcomes" [44]. We hypothesized that the interventions would affect household norms and economic assets, and adolescent female educational, health, social, and economic assets. The theory of change highlights how these potential short-term benefits, important in their own right, are also mediating factors for the longer-term primary objective of AGI-K: delayed childbearing. Because premarital sex is common in this population (S1 Table), we hypothesized that delayed childbearing would result from delayed sexual debut and/or increased family planning and contraceptive use rather than, for example, delayed marriage.

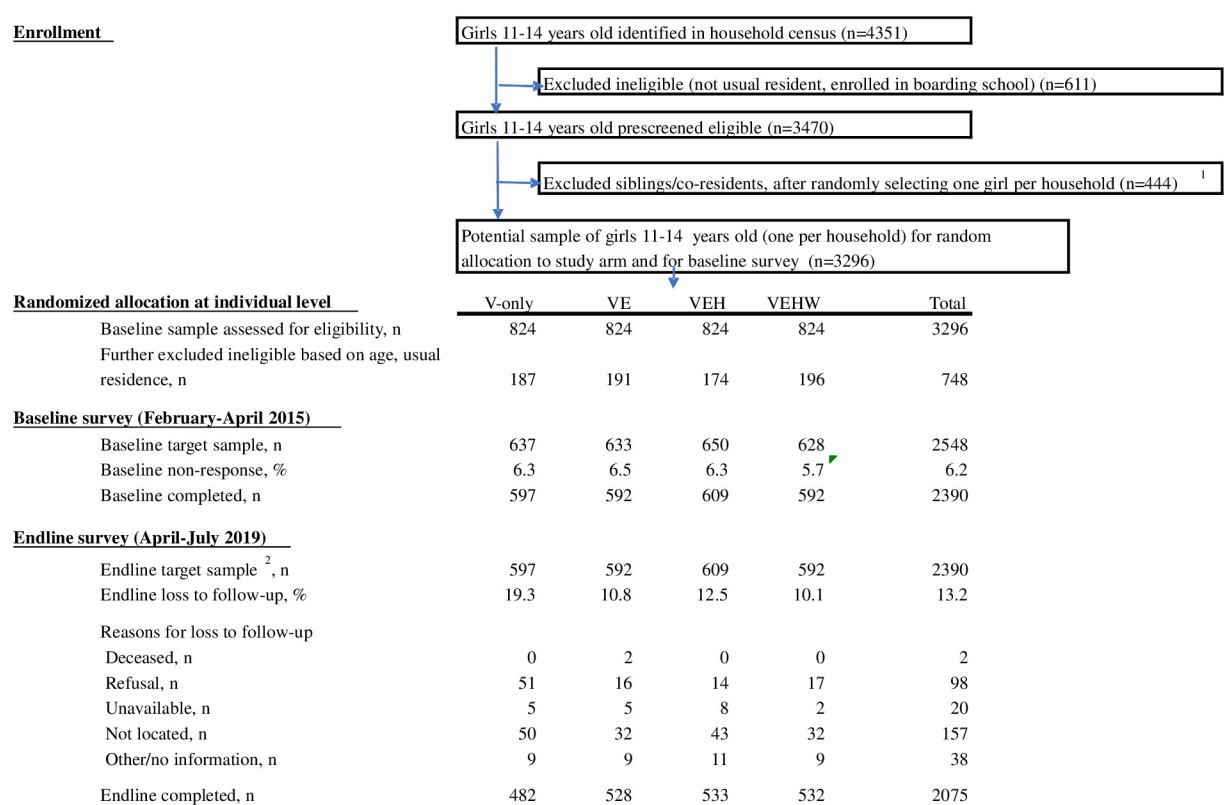

**Fig 1. Individual sample flow including reasons for loss to follow-up.** [1]If the first random selected girl was later determined ineligible or was unavailable, an eligible sister was substituted in her place in the baseline survey. [2]The baseline completed sample was the target sample at endline.

## Interventions

The AGI-K program packages included nested combinations of four single sector-specific interventions. The four sector-specific interventions are described in Fig 3 and included violence prevention through community conversations (CC) [45] to address sexual and physical violence and the devaluation of girls and women; an education intervention comprised of cash and in-kind transfers conditioned on school enrollment and attendance; health and life skills (HLS) education provided through mentor-led group meetings; and wealth creation including financial education (FE) and savings activities. The theory of change posits complementarities between the different sectors and a common underlying enabling environment in the community. Therefore, rather than examine each single-sector intervention in isolation the study examined the effectiveness of different multisectoral packages of interventions compared with a base intervention addressing violence, implemented throughout the study area so that there was no pure control group in the randomized design (Fig 4). (We return to the implications of this aspect of the study design below in Statistical Methods.) Each intervention component was implemented for two years, from August 2015 through July 2017.

The Population Council-Kenya oversaw the program which was implemented and comprehensively monitored by the nongovernmental organization (NGO) Plan International. We used administrative data collected by the NGO during program monitoring to summarize key indicators of implementation and take-up for each intervention, and to demonstrate that

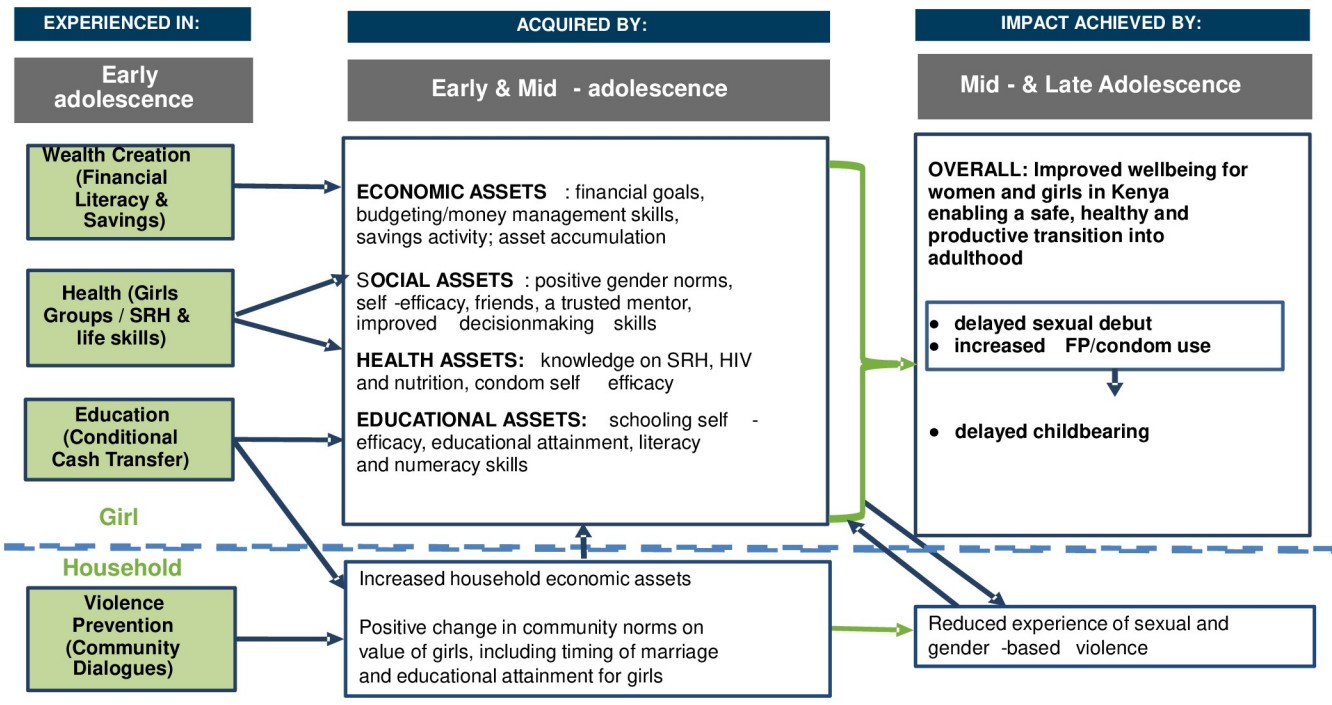

**Fig 2. AGI-K theory of change.** Source: Adapted from Fig 1 open access in Austrian et al. (2016) [40].

program fidelity and participation in the interventions were high. All beneficiary girls were registered in an electronic database managed by the Population Council to which the NGO submitted updates via a mobile phone application. The database included enrollment and attendance information maintained to verify compliance with conditions prior to making cash or in-kind transfers, all of which were also included in the database. A standard operating procedures manual was developed including protocols for various potential situations, for example to resolve problems with cash transfer delivery. In addition, each girls group meeting (including date and time, location, identity of mentor, topics covered, and a list of girls in attendance) was captured in the database. The Population Council and the NGO held a two-week training session for mentors at the start of the program and a one-week refresher session at the midpoint. In addition, periodic observation visits for each mentor and her groups were conducted and a checklist used to verify key indicators of implementation quality. Lastly, monthly review meetings were held with all mentors to identify and address challenges and undertake additional training as needed.

## Randomization and data collection

By design, all community members living in the study area were exposed to the violence prevention intervention since it operated at the community level throughout the study area. For the other interventions (education, health, and wealth creation), the primary targeted beneficiaries were resident girls 11–14 years old at the start of the program. High population density and widespread availability of schools in Kibera [46] made it possible to reach a large number of girls there with different intervention packages. For example, it was feasible to offer VE to some girls in the study area and VEH to others, inviting the latter to participate in the health intervention girls meetings while the former were excluded from that intervention component.

| Intervention | Key Intervention Design Components | Implementation |
|---|---|---|
| Violence prevention | Community conversations (CC) and contracts<br>• Formation of key stakeholder committees<br>• Regular facilitated committee meetings discussing challenges and how to improve conditions for girls (and women) in the community<br>• Development and implementation of action plans (budget allocations ~$2000) | • Committees formed in seven different areas of Kibera<br>• Met 30.0 times on average between December 2015 and July 2017<br>• Action plans completed in all seven areas focused on setting up resource centers or libraries for girls |
| Education | Conditional cash and in-kind transfers<br>• Two cash transfers to household head per term (~$11 per transfer)<br> • 1st transfer upon enrollment<br> • 2nd transfer upon verified continued attendance<br>• In-kind personal and school supplies kit to the girl upon enrollment per term (value ~$6)<br>• Cash transfer paid to school upon enrollment per term partially covering school fees (up to ~$7 for primary and ~$60 for secondary)<br>• Cash transfer incentive paid to school upon enrollment per term (~$5 for each girl) | • 93% of eligible girls received at least one cash transfer<br>• Average 9.5 (of possible 12) transfers received<br>• Average 5.0 (of possible 6) school term fees paid<br>• Average 4.2 (of possible 6) school term supply kits received<br>• No transfers made to girls in V-only study arm |
| Health | Health and life skills (HLS) training[1]<br>• Girls groups led by trained female mentor from the community, intended to provide a safe space for discussion<br>• 20–25 girls per group, segregated by age at baseline (11–12, 13–14 years old)<br>• Meet weekly<br>• Units include: 1) Introductory Sessions; 2) Reproductive Health; 3) Nutrition; 4) Life Skills; 5) HIV, AIDS and STIs; 6) Gender-Based Violence; 7) Harmful Practices; 8) Leadership; 9) Human Rights; 10) Water, Sanitation and Hygiene<br>• 47 sessions in HLS curriculum | • Eligible girls attended 36.0 sessions on average<br>• 80% attended at least 12 meetings<br>• Meetings not offered to girls assigned to V-only or VE study arms |
| Wealth creation | Financial education (FE) and savings activities[1]<br>• Financial education integrated into girls group meetings<br>• 19 sessions in curriculum covering inter alia saving, setting financial goals, budgeting, how to prioritize spending and financial negotiation<br>• Annual savings incentive to girls (~$3)<br>• Facilitated opening formal savings accounts including field trip to local bank | • Eligible girls attended 10.5 financial education (FE) sessions on average<br>• 82% of eligible girls opened bank account<br>• 79% of girls received both annual savings incentives<br>• FE sessions offered only to girls assigned to VEHW study arm |

**Fig 3. AGI-K Kibera intervention design and take-up.** Note: [1]Detailed curricula for HLS and the FE content available at: https://www.popcouncil.org/research/adolescent-girls-initiative-action-research-program.

| Intervention package/Study arm | Abbreviation |
|---|---|
| Violence Prevention Only | V-only |
| Violence Prevention + Education | VE |
| Violence Prevention + Education + Health | VEH |
| Violence Prevention + Education + Health + Wealth Creation | VEHW |

**Fig 4. Intervention packages/study arms.**

Therefore, we implemented an individual-level randomized design in which the unit of randomization was the girl (and her household).

From November 2014 to January 2015 a complete household census of the study area was done to identify all potentially eligible girls. Girls were eligible for the program if they were 1) 11–14 years old; 2) usually residing in the study area; and 3) not currently enrolled in boarding school, since participation in the locally-based girls meetings would be infeasible for those studying away. Some households had more than one potentially eligible girl. A list with one girl from each household (randomly selecting one girl from households with more than one) was prepared and girls (and their households) on the list were assigned to study arms during a public meeting attended by local leaders and other stakeholders. Random assignment to study arm was done publicly to ensure transparency and strengthen program acceptance in the community. A spreadsheet was projected onto a screen with a line for each girl (identified on the spreadsheet only by an anonymous identification number) and a random number was generated for each girl. The list was put in ascending order based on the generated random number and then divided into four equally sized groups. Each group was assigned to a study arm when four community representatives in turn blindly drew a card from a set of four cards, each indicating one of the study arms.

Subsequently a quantitative baseline survey was administered, prior to unblinding of study arm assignment to girls (and their households) and to the start of the program. Baseline enumerators were similarly blinded to the girl's study arm assignment. The baseline survey targeted all girls on the list randomized to study arms, but reconfirmed eligibility prior to carrying out an interview. At a later date, study arm assignment was revealed to the interviewed girl and all other eligible girls in her household, if any, were invited to participate in the program in the same study arm. The baseline survey was conducted February–April 2015. All girls interviewed at baseline were targeted for longitudinal follow-up two years later at the end of the program (May–July 2017) and then four years later at endline (April–July 2019) (Fig 1).

To complement the quantitative surveys, we also carried out a qualitative evaluation of the program in April 2018, about nine months after the program had ended. The purpose of the interviews was to assess the enduring perceptions of the strengths and weaknesses of the interventions and the experience and effects of the interventions on participants, as well as to understand knowledge, attitudes, and practices in the communities. In-depth individual semi-structured interviews were carried out with 28 adolescent girls, eight parents and teachers, four mentors, four CC facilitators, and six community gatekeepers. To facilitate efficient investigation of all four interventions the sample of girls was drawn from beneficiaries in the VEHW study arm and was stratified by number of girls group meetings attended (above and below the median) and whether the girl had opened a bank account. Parents were selected from households with a girl included in the qualitative sample. With the assistance of the NGO, the other adult participants in the qualitative interviews were selected based on their knowledge of and experience with the program.

## Quantitative methodology

**Outcomes.** The primary outcomes include binary 0/1 variables measured at endline and equal to one if the girl had ever: 1) had sex; 2) been pregnant; or 3) given birth.

We also directly examined HSV-2, an important health outcome in its own right and also useful as an objectively measured outcome that can be used to corroborate self-reported sexual activity. Trained HIV services counselors collected biological blood specimens via a finger prick for girls 15 years old and older in 2017 and again at endline in 2019 to test for HSV-2 (S2 Text). For the subgroup of girls with HSV-2 measurements, we examined binary 0/1 variables

equal to one if the girl: 1) tested positive for HSV-2 at endline (i.e., prevalence in 2019); and 2) tested positive for HSV-2 at endline having tested negative in 2017 at the two-year follow-up, identifying girls who seroconverted from negative to positive between 2017 and 2019 (i.e., incidence between 2017 and 2019). Use of contraceptive methods was logically asked only of girls who reported ever having been sexually active (N = 367), therefore statistical analysis of this indicator was not done for the small, highly selective subsample.

We also pre-specified secondary outcomes within each of the four domains reflecting the underlying mechanisms in the theory of change: violence prevention, education, health (particularly sexual and reproductive health [SRH] knowledge), and wealth creation. We present summary measures for the secondary outcome domains at endline, providing results for their component parts in S6 Table; impacts on the secondary outcomes in 2017 at the end of the two-year intervention are reported elsewhere [35]. Variable definitions are provided in S7 Table.

**Sample size and power analysis calculations.** A minimum detectable effect (MDE) approach was used to conduct a power analysis based on the number of potential beneficiaries who could be covered by the program budget and included in the survey, approximately 3,000 across the four study arms. MDEs comparing each of the VE, VEH, or VEHW study arms to the V-only study arm were estimated for prevalence of first birth at endline [40]. Setting the power at 80 percent and the significance level at 5 percent, power analysis was conducted for two-sample proportions tests in the statistical analytical software STATA because each comparison involved two groups, one study arm with a package of interventions (e.g., VE) compared to the V-only study arm. Based on the 2012 Nairobi Cross-Sectional Slum Survey (NCSSS) [36], we assumed that 15.4 percent of girls in the V-only study arm would have given birth by endline when girls in the sample were 15–18 years old. Using individual randomization and an estimated final sample size of 600 girls per arm (3,000/4 = 750 girls per study arm at baseline, assuming a loss to follow-up of 20 percent over the four years) allowed an MDE difference of 5.4 percentage points between the V-only and each of the other study arms. Because of a higher-than-expected proportion of ineligible girls after complete enumeration of the study area, however, the attained baseline sample included approximately 600 girls per arm, which after an assumed rate of 20 percent attrition allowed for an MDE of 6.3 percentage points.

**Statistical methods.** We assessed balance on baseline characteristics across the randomized study arms for the sample reinterviewed at endline to explore potential bias from nonrandom attrition. We also assessed attrition using ordinary least squares to estimate the probability of reinterview at endline and examined whether correlates of attrition differed by study arm.

We then estimated the intent-to-treat (ITT) effect of each package of interventions relative to the V-only arm at endline. Because the violence prevention intervention was included in all study arms, the research was not designed to estimate the impact of that intervention alone. ITT was defined as a girl (and her household) assigned at baseline to a specific study arm, irrespective of her actual participation in the AGI-K programs. Analysis of covariance (ANCOVA) models were used in which the baseline value for the outcome variable, when available, was included as a control.

Because three interrelated primary-outcomes indicators were evaluated, we also combined them into a single summary measure to account for concerns related to multiple hypothesis testing. For each individual primary outcome we calculated a z-score based on the mean and standard deviation (SD) of the V-only study arm at endline. Using those, we constructed an inverse covariance weighted index, restandardizing to be mean 0 and SD 1 [47]. We then estimated the same ITT model on this summary z-score in which coefficient estimates reflect

changes measured in standard deviations. The same methodology was applied to construct summary measures for the secondary outcomes.

All regressions included controls for age and, when available, the baseline value of the outcome measure for the ANCOVA. In addition, per the study protocol [40] we report regressions with additional controls for baseline schooling, cognitive skills, parental characteristics and household wealth to improve precision and account for any initial imbalance [40]. We also examined results combining all three study arms with the education conditional cash transfer (VE, VEH and VEHW) into a single indicator and estimated the average overall effect of those three pooled study arms compared to V-only. We conducted subgroup analysis on girls 13 years old and older at baseline, the subgroup for whom HSV-2 testing was done both at the two-year follow-up in 2017 and again at endline in 2019. Because of their older ages, this subgroup was more likely to have begun having sex, become pregnant or given birth at endline. We consider this analysis post-hoc because it was not outlined in the original study protocol [40]. Finally, as an additional sensitivity analysis, we constructed inverse probability weights (IPW) based on a comprehensive model of attrition, reporting weighted results in the supplementary appendix.

All regressions were estimated with robust standard errors and we set statistical significance at 5 percent. Statistical analysis was conducted using STATA 15.1.

## Qualitative methodology

In order to conduct the qualitative evaluation of the trial, the interviewers and focus group moderators followed semi-structured interview guides which had questions and probes on topics related to AGI-K including individual attitudes on and experiences with the various intervention components; community impressions of and involvement in the program; and community attitudes and norms related to gender, gender-based violence, romantic/sexual relationships, and family planning among adolescent girls. Interview guides were available in English and Swahili, the national languages of Kenya, and the interviews were conducted in the respondent's language of choice. The interviews were conducted by trained interviewers and moderators, and interviewers and respondents were matched by gender.

All interviews were recorded with participant permission and transcribed directly into English. Following transcription, all transcripts were validated and reviewed for quality assurance by a second validator prior to being coded. No personal identifying information, other than the assigned participant identification were included in the transcriptions. An initial starting list of codes was developed and included in a code-book based on the program's theory of change and interview guides, and then additional codes were added as new themes emerged from the data [48]. All transcripts were double-coded by two qualified analysts using ATLAS.ti. To test for intercoder agreement across the double-coded transcripts, a Krippendorff's [49] c-α-binary coefficient was obtained for all key codes. For cases where the coefficient was less than 0.70, side-by-side comparison, clarification, and reconciliation were carried out on the specific coded transcripts.

## Ethical approval

The Population Council Institutional Review Board (IRB) (protocol (p) 661) and the AMREF Ethics and Scientific Review Committee (p143-2014) approved the study in 2014 prior to any contact with participants or enrollment. In addition, all necessary research permits were obtained from the Kenyan National Council for Science, Technology and Innovation (P/18/6952/25330). Written informed consent was obtained from all girls 18 or older; written parental or guardian consent was obtained for girls under 18 years old, with those girls providing

oral assent. Upon completion of a draft study protocol paper in 2015 [40], the unchanged trial was also retrospectively registered in the ISCRTN registry (ISRCTN77455458) as was required for journal submission; the authors confirm that all ongoing and related trials for this intervention are registered.

# Results

## Intervention take-up—Quantitative and qualitative findings

For the targeted adolescent girls, take-up of the educational components of the intervention was high; 90 percent or more of girls randomized to a study arm including the education intervention received at least one household cash transfer, and out of a possible 12 transfers on average girls received 9.5 (SD 3.7) transfers, with median of 11 transfers (Table 1). Transfers of school fee payments and receipt of in-kind supply kits were similarly high relative to the potential maximum. There were no transfers made to girls in the V-only arm. Attendance at the health and life skills sessions was also high. Over 90 percent of girls in the VEH and VEHW study arms attended at least one of the health and life skills sessions. Overall attendance at group meetings was higher among girls in the VEHW arm than in the VEH arm; however because of the design with the same overall number of group meetings, on average girls in the VEH arm had higher exposure to the health and life skills curriculum than girls in the VEHW arm. By design, girls in VE were not included in the group meetings though administrative data indicate that a negligible fraction attended. Of the girls in the VEHW study arm, 92 percent attended at least one FE session and over the course of the intervention, attended on average 10.5 (SD 7.0) group meetings covering FE. More than 75 percent of girls attended at least four FE sessions. By design, group sessions for girls in the VEH arm did not include the FE curriculum, although administrative data indicate that a few FE sessions were offered to a small number of groups in the wrong study arm with approximately 3 percent of girls in VEH

**Table 1. AGI-K Kibera intervention take-up, by study arm.**

|  | V-only | VE | VEH | VEHW | Overall[1] |
|---|---|---|---|---|---|
| Education intervention |  |  |  |  |  |
| Received at least one cash transfer, % | 0.0 | 92.7 | 90.1 | 94.6 | 92.5 |
| Cash transfers received (out of 12), mean | 0.0 | 9.4 | 9.2 | 9.9 | 9.5 |
| School fee payments received (out of 6), mean | 0.0 | 4.9 | 4.9 | 5.0 | 5.0 |
| School kits received (out of 6), mean | 0.0 | 4.1 | 4.1 | 4.4 | 4.2 |
| Health intervention |  |  |  |  |  |
| Total group meetings attended,[2] mean | 0.0 | 0.0 | 34.5 | 37.6 | 36.0 |
| Attended at least 12 group meetings, % | 0.0 | 0.2 | 77.5 | 82.8 | 80.1 |
| Health and life skills sessions attended, mean | 0.0 | 0.0 | 34.3 | 27.0 | 30.7 |
| Wealth-creation intervention |  |  |  |  |  |
| Financial education sessions attended, mean | 0.0 | 0.0 | 0.1 | 10.5 | 10.5 |
| Attended at least 4 financial education sessions, % | 0.0 | 0.0 | 2.1 | 80.7 | 80.7 |
| Received both annual savings incentives, % | 0.0 | 0.0 | 0.0 | 78.7 | 78.7 |
| Opened savings account, % | 0.0 | 0.5 | 0.3 | 81.9 | 81.9 |
| N | 597 | 592 | 609 | 592 | 2,390 |

Source: Program administrative data collected during program monitoring by the implementing NGO.

[1] Overall average across applicable study arms (VEH and VEHW for health intervention and VEHW for wealth-creation intervention).

[2] Groups met weekly over two years for a maximum of ~100 meetings.

apparently exposed to a FE session outside their study arm. All indicators of take-up for girls in the endline analytical sample examined in the paper were similar or slightly higher.

**Community conversation experiences.** Respondents noted that initially there was good attendance at the CC meetings however, this decreased over time partly due to a lack of commitment by some participants in the absence of individual monetary incentives. In addition, other responsibilities made it difficult to always attend the meetings. Some respondents indicated that participants were predominantly female.

*'So. . . in future if you need to meet the parents, you should send a text message via phone. For example, if you need us on Saturday, send us the message by Friday or Thursday so I can plan my work and make time for PLAN* [CC meetings organized by Plan International].'

***Father*** *(unknown age).*

In the qualitative interviews, respondents who participated in the CC violence prevention meetings indicated the meetings covered a range of topics including parenting, challenges facing adolescents, violence resolution, and education. (The qualitative sample is described further below). Participants reported matters of concern to the communities were discussed openly and that they learned useful parenting lessons.

'*Before then, when violence was meted against a girl, the parents did not know how to pursue the cases and they used to be paid something small by the perpetrators and they kept silence. At least right now people are informed that they know what to do to fight people who violate girls, that voice can be heard in the community.'*

***CC facilitator, male*** *(unknown age).*

## Intervention effects—Quantitative and qualitative findings

Fig 1 presents the detailed sample flow by study arm. After complete enumeration of the study areas, there were fewer eligible girls than the 3,000 anticipated in the design. At baseline assessment for eligibility, the target baseline sample for interview was 2,548 girls, of whom 2,390 were interviewed, with similar rates of nonresponse (5.7–6.5 percent) across study arms (prior to informing individuals of treatment status). Because there was substantial residential mobility for girls, to keep attrition to a minimum we implemented brief periodic tracking surveys, updating location and contact information between comprehensive survey rounds. In April–July 2019, 2,075 (86.8 percent) baseline girls were reinterviewed with rates differing across study arms (80.7–89.9 percent). Despite tracking girls to 31 of 47 counties throughout Kenya, one common reason for loss to follow-up was not being able to locate girls who had moved; a second common reason, particularly in the V-only study arm, was refusal.

Table 2 presents means at the start of the program for the sample of girls reinterviewed at endline. At baseline, the girls averaged 12.6 years old and about half lived with their parents. Two-thirds of mothers and three-quarters of fathers had themselves completed primary school. Almost one-third of girls had experienced violence by a male in the past year. Virtually all were enrolled in school, over 90 percent were literate, and average grade attainment was 5.7 years. SRH knowledge at baseline was low and less than 2 percent of girls reported having had sex, been pregnant, or given birth (S2 Table). The endline samples were balanced on a range of baseline characteristics across study arms, with no large or statistically significant differences. S2 Table reports means by study arm for the full baseline sample, and shows similar patterns [50].

**Table 2. Baseline means for endline analytical sample, by study arm.**

| | (1) | (2) | (3) | (4) | (5) |
|---|---|---|---|---|---|
| | V-only | VE | VEH | VEHW | p-value |
| Background | | | | | |
| Age in years, mean (SD) | 12.6 (1.2) | 12.5 (1.3) | 12.6 (1.3) | 12.5 (1.2) | 0.573 |
| Cognitive score (0–16), mean (SD) [n = 2,059] | 8.2 (3.1) | 8.4 (3.0) | 8.4 (3.2) | 8.3 (3.1) | 0.614 |
| Lives with both parents (= 1), % [n = 2,058] | 51.7 | 57.0 | 51.1 | 53.9 | 0.219 |
| Mother completed primary school (= 1), % [n = 1,934] | 63.6 | 64.0 | 62.8 | 64.4 | 0.962 |
| Father completed primary school (= 1), % [n = 1,714] | 76.2 | 79.6 | 74.9 | 79.1 | 0.274 |
| Violence prevention | | | | | |
| Experienced violence by a male in the past year (= 1), % | 29.5 | 29.5 | 30.2 | 32.0 | 0.807 |
| Positive gender attitudes score (0–4), mean (SD) | 3.6 (0.7) | 3.6 (0.7) | 3.6 (0.7) | 3.6 (0.7) | 0.820 |
| Education | | | | | |
| Grade attainment, mean (SD) | 5.7 (1.4) | 5.7 (1.3) | 5.7 (1.3) | 5.6 (1.4) | 0.659 |
| Primary school complete (= 1), % | 7.9 | 5.9 | 6.9 | 6.0 | 0.570 |
| Enrolled in school (= 1), % | 99.2 | 99.1 | 98.3 | 99.2 | 0.550 |
| Literate in Swahili and English (= 1), % [n = 2,059] | 91.9 | 94.1 | 94.1 | 94.1 | 0.448 |
| Health | | | | | |
| Knows most fertile period during menstrual cycle (= 1), % | 8.3 | 7.6 | 7.1 | 5.8 | 0.450 |
| General self-efficacy score (0–6), mean (SD) | 3.9 (1.7) | 4.0 (1.6) | 4.0 (1.6) | 3.9 (1.6) | 0.735 |
| Wealth creation | | | | | |
| Financial literacy score (0–10), mean (SD) [n = 2,014] | 5.7 (1.9) | 5.7 (1.9) | 5.7 (1.9) | 5.7 (1.8) | 0.946 |
| Saved money in the past six months (= 1), % [n = 2,014] | 27.6 | 25.2 | 25.9 | 29.1 | 0.475 |
| Worked for income in the last year (= 1), % | 10.6 | 12.3 | 10.9 | 10.3 | 0.756 |
| Household-level | | | | | |
| Household expects girl to complete secondary (= 1), % [n = 2,117] | 99.8 | 99.6 | 99.8 | 100.0 | 0.261 |
| Household wealth quintile (1–5), mean (SD) [n = 2,132] | 3.1 (1.4) | 3.1 (1.4) | 3.0 (1.4) | 3.0 (1.4) | 0.395 |
| Sample by arm when n = 2,075 | 482 | 528 | 533 | 532 | |

Notes: P-values in column 5 are from an F-test for joint differences across study arms for sample of nonattritors at endline. N = 2,075 unless otherwise noted. Sample sizes are larger for household-level variables because for some observations the household but not the individual survey was completed. All statistical tests were carried out using robust standard errors.

*** p<0.001,

** p<0.01,

* p<0.05,

† p<0.1.

Linear probability models predicting reinterview at endline are presented in S3 Table. The probability of reinterview was lower for the V-only study arm and higher for the youngest girls, those with higher completed grades, and those who resided with both parents. Expanding the model to include interactions of the controls with an indicator for each study arm, there were no significant differences in the relationships between the covariates and attrition in each study arm. This pattern, alongside the evidence of balance across study arms after attrition (Table 2), suggests that although there was measurable attrition, large systematic biases threatening internal validity are unlikely.

Table 3 presents results from the ITT analyses on the primary outcomes for the full sample in the top panel and for subgroup analysis of girls 13 years old and older at baseline in the bottom panel. At endline in the V-only study arm for the full sample, 21.0 percent of girls reported

**Table 3. Estimated intent-to-treat effects on primary outcomes at endline, by study arm.**

| | (1) | (2) | (3) | (4) | (5) | (6) | (7) | (8) | (9) |
|---|---|---|---|---|---|---|---|---|---|
| | V-only endline mean | VE estimate | VEH estimate | VEHW estimate | VE-VEH-VEHW pooled estimate | VE estimate: extended controls | VEH estimate: extended controls | VEHW estimate: extended controls | VE-VEH-VEHW Pooled estimate: extended controls |
| **Full sample** | | | | | | | | | |
| Ever had sex (= 1) | 0.210 | -0.033 | -0.037 | -0.032 | -0.034† | -0.031 | -0.042† | -0.031 | -0.034† |
| 95% CI | | [-0.08, 0.01] | [-0.08, 0.01] | [-0.08, 0.01] | [-0.07, 0.00] | [-0.08, 0.01] | [-0.09, 0.00] | [-0.08, 0.01] | [-0.07, 0.00] |
| Ever pregnant (= 1) | 0.077 | -0.014 | 0.006 | -0.018 | -0.009 | -0.014 | 0.002 | -0.019 | -0.010 |
| 95% CI | | [-0.04, 0.02] | [-0.03, 0.04] | [-0.05, 0.01] | [-0.03, 0.02] | [-0.04, 0.02] | [-0.03, 0.03] | [-0.05, 0.01] | [-0.04, 0.02] |
| Ever given birth (= 1) | 0.066 | -0.023† | 0.007 | -0.023† | -0.013 | -0.023† | 0.002 | -0.024† | -0.015 |
| 95% CI | | [-0.05, 0.00] | [-0.02, 0.04] | [-0.05, 0.00] | [-0.04, 0.01] | [-0.05, 0.00] | [-0.03, 0.03] | [-0.05, 0.00] | [-0.04, 0.01] |
| *Fertility outcomes summary index z-score* | 0.000 | -0.090 | -0.035 | -0.092† | -0.072 | -0.086 | -0.048 | -0.090† | -0.075 |
| 95% CI | | [-0.20, 0.02] | [-0.15, 0.08] | [-0.20, 0.01] | [-0.16  0.02] | [-0.19, 0.02] | [-0.16, 0.07] | [-0.19, 0.01] | [-0.17, 0.02] |
| **Baseline 13–14-year-olds [n = 1,007]** | | | | | | | | | |
| Ever had sex (= 1) | 0.325 | -0.082* | -0.062 | -0.063 | -0.069* | -0.085* | -0.071† | -0.061 | -0.072* |
| 95% CI | | [-0.16, -0.01] | [-0.14, 0.01] | [-0.14, 0.01] | [-0.13, 0.00] | [-0.16, -0.01] | [-0.15, 0.00] | [-0.14, 0.01] | [-0.14, -0.01] |
| Ever pregnant (= 1) | 0.118 | -0.023 | 0.016 | -0.018 | -0.008 | -0.022 | 0.013 | -0.015 | -0.008 |
| 95% CI | | [-0.08, 0.03] | [-0.04, 0.07] | [-0.07, 0.04] | [-0.05, 0.04] | [-0.07, 0.03] | [-0.04, 0.07] | [-0.07, 0.04] | [-0.05, 0.04] |
| Ever given birth (= 1) | 0.101 | -0.026 | 0.013 | -0.024 | -0.012 | -0.026 | 0.008 | -0.023 | -0.014 |
| 95% CI | | [-0.07, 0.02] | [-0.04, 0.07] | [-0.07, 0.02] | [-0.05, 0.03] | [-0.07, 0.02] | [-0.04, 0.06] | [-0.07, 0.02] | [-0.05, 0.03] |
| *Fertility outcomes summary index z-score* | 0.000 | -0.136† | -0.047 | -0.111 | -0.098 | -0.139† | -0.064 | -0.106 | -0.103 |
| 95% CI | | [-0.29, 0.02] | [-0.22, 0.12] | [-0.27, 0.05] | [-0.23, 0.04] | [-0.30, 0.02] | [-0.23, 0.10] | [-0.26, 0.05] | [-0.23, 0.03] |
| HSV-2 positive[1] (= 1) [n = 938] | 0.204 | -0.075* | 0.029 | -0.043 | -0.029 | -0.072* | 0.033 | -0.041 | -0.026 |
| 95% CI | | [-0.14, -0.01] | [-0.05, 0.10] | [-0.11, 0.03] | [-0.09, 0.03] | [-0.14, 0.00] | [-0.04, 0.11] | [-0.11, 0.03] | [-0.09, 0.03] |
| HSV-2 incidence 2017-19[2] (= 1) [n = 740] | 0.091 | -0.056* | -0.007 | -0.023 | -0.029 | -0.055* | -0.005 | -0.022 | -0.028 |
| 95% CI | | [-0.11, 0.00] | [-0.07, 0.05] | [-0.08, 0.03] | [-0.08, 0.02] | [-0.11, 0.00] | [-0.07, 0.05] | [-0.08, 0.04] | [-0.08, 0.02] |

Notes: Sample is N = 2,075 unless otherwise indicated. Endline means for the V-only study arm are reported in column 1 and the estimated ITT effect for each study arm relative to V-only in columns 2–4. Column 5 pools all three intervention arms with education into a single treatment indicator. Columns 6–9 report estimated ITT effects with extended controls. Numbers in square brackets indicate 95% confidence intervals. Regressions were estimated with robust standard errors and included controls for age and the outcome measured at baseline. The extended control regressions additionally control for baseline measures of: grade attainment, cognitive score, mother or father completing primary school, coresidence with both parents, household wealth quintile, and whether any missing baseline covariates were imputed using area median. [1] No baseline control for outcome variable available. [2] Among respondents who tested HSV-2 negative in 2017.

*** p<0.001,

** p<0.01,

* p<0.05,

† p<0.1.

ever having had sex, 7.7 percent had ever been pregnant, and 6.6 percent had ever given birth. The majority of the estimated ITT effects (compared to V-only) on the primary outcomes and summary measure in columns 2–4 are negative. However, only the effects for ever having given birth in the VE and VEHW study arms are significant at 10 percent. Magnitudes of the estimated ITT effects with extended controls are similar. There was a nearly 0.1 SD reduction in the fertility outcomes summary measure for the VEHW study arm, also significant only at 10 percent. Estimates accounting for further potential attrition bias by reweighting with IPW (S1 Text) indicated similar reductions for the summary fertility indicator in both the VE and VEHW study arms (S4 Table), consistent with the evidence on balance across study arms after attrition.

For the subgroup of girls 13 years and older at baseline who were measured at endline in the V-only arm, 32.5 percent reported having ever had sex, 11.8 percent had been pregnant, and 10.1 percent had given birth. Prevalence of HSV-2 among these girls was 20.4 percent, with 9.1 percent of the girls who tested negative in 2017 seroconverting and testing positive by 2019. Estimated effects on ever having had sex and prevalence and incidence of HSV-2 were negative and statistically significant in the VE study arm.

Turning to the secondary outcomes (Table 4), as expected given the experimental design (in which all study arms received the violence prevention intervention), compared to V-only there were no significant effects on the violence prevention summary measures. There were increases in education of 0.1 SD or more, largest for the VE study arm but smaller and significant only at the 10 percent level for the other, more complex study arms in which girls had more program responsibilities. Study arms with the health intervention all increased the health outcome summary score and VEHW had a large effect on wealth creation. Pooling the three study arms with the education component into a single indicator yields clear evidence of positive impacts on the education, health, and wealth summary outcomes of between 0.1 and 0.2 SD. The effects on secondary outcomes were similar or stronger than the effects estimated at the two-year follow-up. For the sample of girls 13 years old and older at baseline, effects were similar for all but the health outcomes summary where effects were more muted, possibly because SRH had increased across arms for all the girls over time. Results for secondary outcomes were robust to weighting for attrition, with point estimates and significance levels changing only slightly (S5 Table). S6 Table presents results for each individual component of the secondary summary measure outcomes. Notably, knowledge of at least one form of modern contraception increased 5 percentage points or more in VE, VEH, and VEHW, and condom self-efficacy increased by 0.13 SD in VEH.

The sample of participants interviewed in the qualitative evaluation (discussed above) is shown in Table 5. All adolescent girls interviewed were under 18 years old, 68 percent (n = 19) were in secondary school, none had been married, and around 80 percent (n = 22) were Protestant. Girls were selected to represent different levels of group session participation and access to a personal bank account. Of the parents who were interviewed, 36 percent (n = 3) were mothers. Gatekeepers included religious leaders and chiefs within the community. Below we explore the qualitative findings from the subthemes that were identified from the core themes of education, schooling, sexual behavior, and marriage.

**Barriers to education.**   The reported barriers to education included lack of school fees, pregnancy, peer pressure, child sexual exploitation, and drug abuse. Respondents suggested several potential ways to overcome these barriers and improve learning outcomes which included paying school fees on time, providing materials such as books, hiring more teachers, and empowering girls.

**Table 4. Estimated intent-to-treat effects on secondary outcomes summary measures at endline, by study arm.**

| | (1) | (2) | (3) | (4) | (5) | (6) | (7) | (8) |
|---|---|---|---|---|---|---|---|---|
| | VE estimate | VEH estimate | VEHW estimate | VE-VEH-VEHW pooled estimate | VE estimate: extended controls | VEH estimate: extended controls | VEHW estimate: extended controls | VE-VEH-VEHW pooled estimate: extended controls |
| **Full sample** | | | | | | | | |
| *Violence prevention outcomes summary index z-score* | 0.008 | 0.015 | 0.003 | 0.009 | -0.013 | 0.004 | 0.002 | -0.002 |
| 95% CI | [-0.12, 0.13] | [-0.11, 0.14] | [-0.12, 0.13] | [-0.09, 0.11] | [-0.13, 0.11] | [-0.12, 0.12] | [-0.12, 0.13] | [-0.10, 0.10] |
| *Education outcomes summary index z-score* | 0.175** | 0.096† | 0.108† | 0.126** | 0.146** | 0.090 | 0.100† | 0.112* |
| 95% CI | [0.07, 0.28] | [-0.02, 0.21] | [0.00, 0.22] | [0.03, 0.22] | [0.04, 0.25] | [-0.02, 0.20] | [-0.01, 0.21] | [0.02, 0.20] |
| *Health outcomes summary index z-score* | 0.095 | 0.187** | 0.126* | 0.136** | 0.069 | 0.167** | 0.119† | 0.119* |
| 95% CI | [-0.03, 0.22] | [0.07, 0.31] | [0.00, 0.25] | [0.04, 0.24] | [-0.05, 0.19] | [0.05, 0.28] | [0.00, 0.24] | [0.02, 0.22] |
| *Wealth creation outcomes summary index z-score* | 0.158* | 0.108† | 0.407*** | 0.225*** | 0.128* | 0.093 | 0.407*** | 0.210*** |
| 95% CI | [0.04, 0.28] | [-0.02, 0.23] | [0.29, 0.53] | [0.12, 0.33] | [0.01, 0.25] | [-0.03, 0.21] | [0.29, 0.52] | [0.11, 0.31] |
| **Baseline 13–14-year-olds [n = 1,007]** | | | | | | | | |
| *Violence prevention outcomes summary index z-score* | 0.002 | -0.041 | 0.012 | -0.009 | -0.032 | -0.071 | -0.002 | -0.036 |
| 95% CI | [-0.17, 0.17] | [-0.21, 0.13] | [-0.17, 0.19] | [-0.15, 0.13] | [-0.20, 0.13] | [-0.24, 0.10] | [-0.18, 0.17] | [-0.17, 0.10] |
| *Education outcomes summary index z-score* | 0.243** | 0.116 | 0.159* | 0.172* | 0.220** | 0.105 | 0.139† | 0.154* |
| 95% CI | [0.09, 0.40] | [-0.05, 0.28] | [0.00, 0.32] | [0.04, 0.31] | [0.07, 0.37] | [-0.06, 0.27] | [-0.01, 0.29] | [0.02, 0.28] |
| *Health outcomes summary index z-score* | 0.040 | 0.183* | 0.161† | 0.128† | -0.012 | 0.141 | 0.117 | 0.082 |
| 95% CI | [-0.13, 0.21] | [0.01, 0.36] | [-0.03, 0.35] | [-0.02, 0.27] | [-0.18, 0.16] | [-0.03, 0.31] | [-0.07, 0.30] | [-0.06, 0.22] |
| *Wealth creation outcomes summary index z-score* | 0.198* | 0.114 | 0.367*** | 0.225** | 0.157† | 0.079 | 0.343*** | 0.193** |
| 95% CI | [0.02, 0.37] | [-0.06, 0.29] | [0.20, 0.54] | [0.08, 0.37] | [-0.02, 0.33] | [-0.10, 0.25] | [0.18, 0.51] | [0.05, 0.33] |

Notes: Sample is N = 2,075 unless otherwise indicated. The table reports the estimated ITT effect for each study arm relative to the V-only study arm measured in z-scores in columns 1–3. Column 4 pools all three intervention arms with education into a single treatment indicator. Columns 5–8 report estimated ITT effects with extended controls. Numbers in square brackets indicate 95% confidence intervals. Regressions were estimated with robust standard errors and included controls for age and the outcome measured at baseline. The extended controls regressions additionally control for baseline measures of: grade attainment, cognitive score, mother or father completing primary school, coresidence with both parents, household wealth quintile, and whether any missing baseline covariates were imputed (using area median). Z-scores are calculated separately for the full and baseline 13–14 year-old samples.

*** $p < 0.001$,

** $p < 0.01$,

* $p < 0.05$,

† $p < 0.1$.

**Table 5. Socio-demographic characteristics of participants in the qualitative interviews.**

| Variables | Adolescents (n(%)) (N = 28) | Parents (n(%)) (N = 8) | Mentors (n(%)) (N = 4) | Teachers (n(%)) (N = 8) | CC facilitators (n(%)) (N = 4) | Gatekeepers (n(%)) (N = 6) | Totals (n(%)) (N = 58) |
|---|---|---|---|---|---|---|---|
| *Sex* | | | | | | | |
| Male | 0 (0) | 5 (62.5) | 0 (0) | 4 (50) | 4 (100) | 4 (66.7) | 17 (29.3) |
| Female | 28 (100) | 3 (37.5) | 4 (100) | 4 (50) | 0 (0) | 2 (33.3) | 41 (70.7) |
| *Age* | | | | | | | |
| Below 18 | 28 (100) | 0 (0) | 0 (0) | 0 (0) | 0 (0) | 0 (0) | 28 (48.3) |
| 18--29 | 0 (0) | 0 (0) | 4 (100) | 1 (12.5) | 1 (25) | 0 (0) | 6 (10.3) |
| 30–39 | 0 (0) | 2 (25) | 0 (0) | 4 (50) | 3 (75) | 1 (16.7) | 10 (17.2) |
| 40–49 | 0 (0) | 5 (62.5) | 0 (0) | 2 (25) | 0 (0) | 2 (33.3) | 9 (15.5) |
| 50+ | 0 (0) | 1 (12.5) | 0 (0) | 1 (12.5) | 0 (0) | 3 (50) | 5 (8.6) |
| *Education* | | | | | | | |
| Primary | 9 (32.1) | 4 (50) | 0 (0) | 0 (0) | 0 (0) | 0 (0) | 13 (22.4) |
| Secondary | 19 (67.9) | 4 (50) | 3 (75) | 0 (0) | 0 (0) | 4 (66.7) | 30 (51.7) |
| University/ college | 0 (0) | 0 (0) | 1 (25) | 8 (100) | 4 (100) | 2 (33.3) | 15 (25.9) |
| *Marital status* | | | | | | | |
| Never married | 28 (100) | 0 (0) | 1 (25) | 0 (0) | 0 (0) | 0 (0) | 29 (50) |
| Married | 0 (0) | 6 (75) | 2 (50) | 8 (100) | 4 (100) | 6 (100) | 26 (44.8) |
| Separated | 0 (0) | 2 (25) | 1 (25) | 0 (0) | 0 (0) | 0 (0) | 03 (5.2) |
| *Religion* | | | | | | | |
| Catholic | 3 (10.7) | 0 (0) | 2 (50) | 5 (62.5) | 1 (25) | 2 (33.3) | 13 (22.4) |
| Protestant | 22 (78.6) | 7 (87.5) | 2 (50) | 3 (27.5) | 3 (75) | 4 (66.7) | 41 (70.7) |
| Muslim | 3 (10.7) | 1 (12.5) | 0 (0) | 0 (0) | 0 (0) | 0 (0) | 4 (6.9) |
| Other | 0 (0) | 0 (0) | 0 (0) | 0 (0) | 0 (0) | 0 (0) | 0 (0) |
| *Occupation* | | | | | | | |
| Student | 28 (100) | 0 (0) | 1 (25) | 0 (0) | 0 (0) | 0 (0) | 29 (50) |
| Teacher | 0 (0) | 0 (0) | 0 (0) | 8 (100) | 3 (75) | 0 (0) | 11 (19) |
| Business | 0 (0) | 2 (25) | 1 (25) | 0 (0) | 0 (0) | 3 (50) | 06 (10.3) |
| Other | 0 (0) | 6 (75) | 2 (50) | 0 (0) | 1 (25) | 3 (50) | 12 (20.7) |

*"First, it's the issue of school fees. It should get the student in school the right time. Then, also, you know for some girls, if you haven't understood something, you fear asking. You fear the teacher. So, they should speak."*

**Adolescent, female** (unknown age).

*'The problems that they face at home are like rape, child labor, peer pressure from their fellow age mates, and mostly the biggest problem is the poverty thing. The poverty streamline is on a very high standard and you find that these children undergo a lot, and by doing that it might hinder them from going to school.'*

**Teacher, male, 39**.

**Educational support experiences.** Respondents reported that the effects of the education conditional cash transfer included increased attendance at school and fewer girls being sent home for not having paid school fees. Families benefited from the household transfer and were

able to buy food and clothes for the entire family. The program increased the girls' own educational goals as well as their parents' goals for them. Many girls indicated they were more motivated to study and that their goal of attending university had strengthened. Furthermore, respondents indicated that school performance improved as a result of the program.

*'The part where they used to pay fees. . .you'd learn knowing your fees had been paid for by AGI-K at least and mum could add the other. So, your work was just learning and putting in the effort. That is something that it has helped me with.'*

**Adolescent, female** *(unknown age)*.

**Girls group meeting experiences.** Many girls reported learning important lessons during the girls group meetings, for example on how to maintain hygiene and cleanliness, eat a balanced diet, recognize and protect themselves from sexual and gender-based violence, and delay having sex. Parents and teachers noticed that girls were more open toward them and had greater confidence. Girls also reported enjoying learning how to save money and correspondingly parents observed their increased money management skills and ability to save.

*'What I have seen is good, let's say like now we are depositing for her money, and she is now in form 3. When she reaches form 4, we will continue depositing. So, we are praying to God, that when she completes form four. . .we will look for a college for her to enroll or if it's a university that she will go to we will contribute with that amount so she can enroll with it.'*

*Mother (unknown age).*

Many respondents commented on the various barriers they faced in attending group meetings regularly, including weekend school sessions; travel to their ancestral villages to visit extended family; and lack of parental understanding, support, or encouragement.

*'When [girl's name] got to class 8, she could not attend the meetings because she would go to school even on Sundays. I explained to her that [girl's name] would not be coming for the meetings because she had to go for tuition, and they would be beaten up if they didn't go.'*

**Mother** (unknown age).

**Sexual motivation and pregnancy prevention.** Discussion of sexual behavior in the qualitative interviews elaborated on what motivates girls to engage in sexual relationships and on the circumstances of sex, as well as on pregnancy prevention including common reasons for not using contraception. Respondents reported that girls were likely to engage in exploitative sex for the actual or expected gain of money and gifts and because of peer pressure or influence from the media, but also as a mark of maturity and for sexual adventure.

**"***I think that at times it's the anxiety and the need to discover themselves; they want identity, through peer pressure, and the economic challenges that the girls face. For the older men they could be using money and gifts to lure the girls into that kind of relationship. And again, the parents could not be seeing it as a sin because they see that my daughter is bringing sugar, so its confusion and it is a chain of things.***"**

*Community gatekeeper, male, 46.*

Respondents reported that most girls were aware of family planning methods such as male condoms, pills, and implants as well as where to obtain contraceptives. Some of the reasons cited for not using contraceptives during sex included an inability to negotiate male condom use, having unplanned sex, wanting to avoid experiencing side effects of certain family planning methods, and being the victim of sexual violence. Other reasons included a partner promising to marry them if they became pregnant or wanting to have a child.

"*You know maybe a boy may have lied that he will marry you, and then when he gets you pregnant, he runs away or maybe he rejects that the pregnancy is not his.*"

**Adolescent, female, 14**.

"*Those pills have their side effects especially the many pills that have been introduced into the market*; *they have many side effects.*"

**Adolescent, female, 16**.

## Discussion

In this paper, we report on the effects, of the AGI-K interventions in Kibera targeting young adolescent girls in 2019, two years after the program ended. Although significant at only 10 percent, the percent of girls who had ever given birth was one-third lower among those exposed to the education intervention (VE) or to the full range of intervention components (VEHW), compared to the V-only study arm. The girls were still relatively young (16 years old on average) at the time of the endline survey, however, raising the possibility that program impacts could increase further as they transition to young adulthood. Supporting that possibility are the results for the older subsample of girls who were 13 years old or older at baseline (18 years old on average at endline); these girls had lower outcomes on the fertility summary index in the VE study arm, driven largely by delaying sexual debut.

The reduced fertility outcomes in the VE study arm were corroborated by significantly lower HSV-2 prevalence and incidence in that study arm. Because HSV-2 is almost exclusively transmitted through genital-to-genital contact during sex [51] it has been shown to be reliable marker of sexual behavior [52]. Notably, the findings for HSV-2, a one-third reduction in prevalence and almost two-thirds reduction in incidence, have important beneficial health implications in their own right. Findings from the qualitative study identified a variety of reasons why girls engage in exploitative sexual practices, including for the actual or expected gain of money and gifts. It is possible that the cash transfers played a role in reducing girls' vulnerability to such practices.

Most adolescent pregnancy in Kibera occurs outside of marriage. One of the aims of the focus on SRH knowledge and self-efficacy in the health intervention was to promote changes in sexual behavior, by delaying sexual debut as well as by increasing the ability to negotiate for contraceptive use during sex [53]. Both the quantitative and qualitative findings demonstrated that most girls were knowledgeable about modern contraceptive methods. We were unable to provide quantitative evidence on the effects of the intervention on contraceptive use. The qualitative findings, however, revealed that despite knowledge of methods there remain limitations to use due to girls' inability to negotiate condom use, unplanned sex, fear of experiencing side effects, or wanting to become pregnant. Incomplete contraceptive use in this young sample is consistent with patterns observed across sub-Saharan Africa where women 15–24 years old are found to be more likely to start using contraception after their first birth [54].

Consideration of potential underlying mechanisms, the secondary outcomes in our study, also revealed a number of findings two years after the program ended. Perhaps most important was that the effects were similar to or greater than those estimated at the end of the programs, indicating persistent effects following the early adolescent interventions. More specifically, positive effects were observed in the education outcomes summary index when compared to the V-only study arm. In particular, the pooled study arms estimate showed a significant increase in grade attainment. School enrollment was already high for this age group, likely explaining why there was only minimal impact observed for the other quantitative education indicators such as enrollment. These findings are consistent with qualitative results that identified lack of school fees as a barrier to education. Respondents in the qualitative evaluation reported that the cash transfers increased school attendance, motivation to study and ambition to attend university, as well as reduced the number of girls sent home for unpaid fees.

Persistent positive impacts were similarly observed in the health outcomes summary index for girls in a study arm with the health intervention (VEH and VEHW). Findings from the qualitative survey support the quantitative effects with girls reporting increased SRH knowledge. Parents and teachers also reported positive behavioral change including girls becoming more confident and being more open.

Persistent positive impacts also were observed in the wealth-creation domain, concentrated in the study arm with the wealth intervention. Girls in the VEHW arm had significantly and substantially higher financial literacy scores and were more likely to have saved money, demonstrating that knowledge conveyed in the financial literacy sessions translated into positive behavior change. In line with these observations, the qualitative survey revealed that girls enjoyed learning how to save money and were supported by their parents to do so.

Without a pure control group receiving no intervention, our research design does not identify the impact of the violence prevention study arm alone. Results examining the effect of the other study arms on violence prevention outcomes, however, indicate that there were no effects above and beyond any possible impacts of the violence prevention intervention itself. Qualitative findings highlighted some challenges experienced in implementing the intervention, including low attendance at CC meetings; hence, although resources were delivered and plans completed, it is unclear how efficacious the intervention was.

The theory of change identifies the potential intermediate benefits from the interventions on household norms and economic assets, and on adolescent female educational, health, social, and economic assets, as mediating factors for the longer-term primary objective of delayed childbearing. Earlier work demonstrated that at the end of the intervention in 2017, girls exposed to the VEHW arm had positive effects on grade attainment and large impacts on completion of primary school and transition to secondary school, as well as improved financial literacy and savings behavior. In addition, girls exposed to the health component had improved SRH knowledge and condom-self efficacy [35]. The current study shows that not only did the effects on these mediating factors persist two years after the end of the interventions, but that there was also evidence of longer-term impacts on delayed sexual debut and marginal decreases in adolescent pregnancy.

There are several limitations to this study. First, it was not feasible to implement a full factorial design in which we could also have assessed, for example, the impact of the girls' empowerment groups alone. Second, we are unable to generalize the findings to settings beyond urban informal settlements. Third, the baseline and final quantitative samples were smaller than the target sample because of a lower-than-expected population of eligible girls, thus decreasing power and increasing the MDEs. Fourth, there was a possibility of internal spillover of resources or knowledge to girls in the V-only study arm from girls in other arms, possibly reducing estimated program impacts. Finally, we were unable to evaluate the pathway of

increased contraceptive use in delaying childbearing because of the relatively small sample of girls reporting ever having sex by endline.

At the same time, the study has significant strengths. It is one of few studies using a randomized, longer-term, longitudinal design to examine multisectoral cash plus interventions in a marginalized population. Such follow-up is essential for understanding program impacts on adolescents since early investments may only pay dividends years later, well after the interventions have ended [55]. Fidelity and take-up of the programs were high and there was virtually no program contamination. The study employed a rigorous randomized design with minimal attrition. We were able to corroborate results for self-reported sexual behavior with results for HSV-2 prevalence and incidence, more reliable markers of sexual activity. And last, it was possible to triangulate the quantitative results with qualitative findings from the same sample.

Results from this study demonstrate the potential beneficial effects of multisectoral cash plus interventions consisting of violence prevention and education interventions that target the community and household level, combined with interventions in the education, health, and wealth-creation sectors that directly target individual girls in early adolescence. The study finds that girls exposed to such interventions progress farther in school, have greater SRH knowledge and have improved financial savings behavior—all protective factors against early pregnancy during adolescence. Such interventions, therefore, are likely to have beneficial impacts on the longer-term health and economic outcomes of girls residing in impoverished settings. The study adds to the evidence on whether and how multisectoral interventions that include adolescent-friendly services are able to reduce sex and pregnancy for girls. Future research should be carried out on understanding the impact of short-term cash-plus multisectoral interventions implemented in early adolescence on outcomes that become increasingly common in later adolescence and early adulthood, such as contraceptive use, secondary school completion and income generation.

## Supporting information

**S1 Table. Key indicators for AGI-K primary and secondary outcomes.**
(DOCX)

**S2 Table. Baseline means for full baseline sample, by study arm.**
(DOCX)

**S3 Table. Baseline correlates of endline survey response, by study arm.**
(DOCX)

**S4 Table. Additional estimated ITT effects on primary outcomes at endline, by study arm.**
(DOCX)

**S5 Table. Additional estimated ITT effects on secondary outcomes summary measures at endline, by study arm.**
(DOCX)

**S6 Table. Estimated ITT effects on individual components of secondary outcomes at endline, by study arm.**
(DOCX)

**S7 Table. Endline outcome variable definitions.**
(DOCX)

**S1 Text. Note on attrition weight construction and baseline correlates of endline survey response, by study arm.**
(DOCX)

**S2 Text. HSV-2 testing procedures.**
(DOCX)

**S1 File.**
(DOCX)

**S2 File. CONSORT 2010 checklist of information to include when reporting a randomised trial**\*.
(DOC)

**S3 File.**
(DOCX)

## Acknowledgments

The authors thank our implementing partners at Plan International in Kenya for the high-quality program implementation and strict adherence to the study protocols. We also thank Elizabeth Friesen who provided excellent assistance with data preparation, as well as Population Council staff and data enumerators for implementing data collection. We gratefully acknowledge the efforts of others who contributed to the development of the study design and research instruments, including Eunice Muthengi, Caroline Kabiru, and Joyce Mumah. Finally, we thank all of the adolescent girls and their households who agreed to participate in this study and share their information with us. We also thank Rebecca Balasa, Joyce Wamoyi, and an anonymous referee for valuable comments. All errors and omissions are our own.

## Author Contributions

**Conceptualization:** Karen Austrian, Erica Soler-Hampejsek, Benta Abuya, John A. Maluccio.

**Formal analysis:** Erica Soler-Hampejsek, John A. Maluccio.

**Funding acquisition:** John A. Maluccio.

**Methodology:** Erica Soler-Hampejsek, John A. Maluccio.

**Project administration:** Beth Kangwana, Karen Austrian, Nicole Maddox, Yohannes Dibaba Wado, Benta Abuya, Eva Muluve, Faith Mbushi, Joy Koech.

**Supervision:** Karen Austrian, John A. Maluccio.

**Validation:** Karen Austrian, Erica Soler-Hampejsek, John A. Maluccio.

**Visualization:** Karen Austrian, Erica Soler-Hampejsek.

**Writing – original draft:** Beth Kangwana, Karen Austrian, Erica Soler-Hampejsek, Rachel J. Sapire, John A. Maluccio.

**Writing – review & editing:** Beth Kangwana, Karen Austrian, Erica Soler-Hampejsek, Nicole Maddox, Rachel J. Sapire, Yohannes Dibaba Wado, Benta Abuya, Eva Muluve, Faith Mbushi, Joy Koech, John A. Maluccio.

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
