## [Decision Letter · Decision Letter 0]

6 May 2021

PONE-D-21-06054

Impact of a Cash Transfer and Girls Empowerment Program on Early Sexual Debut and Fertility in a Kenyan Urban Informal Settlement: Results from a Mixed-Methods Randomized Trial

PLOS ONE

Dear Dr. Kangwana,

Thank you for submitting your manuscript to PLOS ONE. After careful consideration, we feel that it has merit but does not fully meet PLOS ONE’s publication criteria as it currently stands. Therefore, we invite you to submit a revised version of the manuscript that addresses the points raised during the review process.

We look forward to receiving your revised manuscript.

Kind regards,

Catherine E Oldenburg

Academic Editor

PLOS ONE

Journal Requirements:

2. Please include the trial registration number in the Abstract.

3. Thank you for submitting your clinical trial to PLOS ONE and for providing the name of the registry and the registration number. The information in the registry entry suggests that your trial was registered after patient recruitment began. PLOS ONE strongly encourages authors to register all trials before recruiting the first participant in a study.

1) your reasons for your delay in registering this study (after enrolment of participants started);

2) confirmation that all related trials are registered by stating: “The authors confirm that all ongoing and related trials for this drug/intervention are registered”.

4. Please include your tables as part of your main manuscript and remove the individual files. Please note that supplementary tables (should remain/ be uploaded) as separate "supporting information" files

5. Please include additional information regarding the survey or questionnaire used in the study and ensure that you have provided sufficient details that others could replicate the analyses. For instance, if you developed a questionnaire as part of this study and it is not under a copyright more restrictive than CC-BY, please include a copy, in both the original language and English, as Supporting Information. In addition, please provide the interview guides used in the qualitative study

6. Thank you for stating the following in the Competing Interests section:

The authors have declared that no competing interests exist.

We note that one or more of the authors are employed by a commercial company: Independent Consultant

6a.              Please provide an amended Funding Statement declaring this commercial affiliation, as well as a statement regarding the Role of Funders in your study. If the funding organization did not play a role in the study design, data collection and analysis, decision to publish, or preparation of the manuscript and only provided financial support in the form of authors' salaries and/or research materials, please review your statements relating to the author contributions, and ensure you have specifically and accurately indicated the role(s) that these authors had in your study. You can update author roles in the Author Contributions section of the online submission form.

6b. Please also provide an updated Competing Interests Statement declaring this commercial affiliation along with any other relevant declarations relating to employment, consultancy, patents, products in development, or marketed products, etc. 

8. We note that Figure 2 in your submission contain copyrighted images. All PLOS content is published under the Creative Commons Attribution License (CC BY 4.0), which means that the manuscript, images, and Supporting Information files will be freely available online, and any third party is permitted to access, download, copy, distribute, and use these materials in any way, even commercially, with proper attribution. For more information, see our copyright guidelines: http://journals.plos.org/plosone/s/licenses-and-copyright.

8a.           You may seek permission from the original copyright holder of Figure(s) [#] to publish the content specifically under the CC BY 4.0 license.

8b.           If you are unable to obtain permission from the original copyright holder to publish these figures under the CC BY 4.0 license or if the copyright holder’s requirements are incompatible with the CC BY 4.0 license, please either i) remove the figure or ii) supply a replacement figure that complies with the CC BY 4.0 license. Please check copyright information on all replacement figures and update the figure caption with source information. If applicable, please specify in the figure caption text when a figure is similar but not identical to the original image and is therefore for illustrative purposes only.

Additional Editor Comments:

Per PLOS ONE guidelines, please explain in the methods why the trial was not prospectively registered. 

Reviewers' comments:

Reviewer's Responses to Questions

**Comments to the Author**

1. Is the manuscript technically sound, and do the data support the conclusions?

Reviewer #1: Partly

Reviewer #2: Partly

Reviewer #3: Yes

2. Has the statistical analysis been performed appropriately and rigorously? 

Reviewer #1: I Don't Know

Reviewer #2: Yes

Reviewer #3: Yes

3. Have the authors made all data underlying the findings in their manuscript fully available?

Reviewer #1: Yes

Reviewer #2: Yes

Reviewer #3: No

4. Is the manuscript presented in an intelligible fashion and written in standard English?

Reviewer #1: No

Reviewer #2: No

Reviewer #3: Yes

5. Review Comments to the Author

Reviewer #1: Important note: This review pertains only to ‘statistical aspects’ of the study and so ‘clinical aspects’ [like medical importance, relevance of the study, ‘clinical significance and implication(s)’ of the whole study, etc.] are to be evaluated [should be assessed] separately/independently. Further please note that any ‘statistical review’ is generally done under the assumption that (such) study specific methodological [as well as execution] issues are perfectly taken care of by the investigator(s). This review is not an exception to that and so does not cover clinical aspects {however, seldom comments are made only if those issues are intimately / scientifically related & intermingle with ‘statistical aspects’ of the study}. Agreed that ‘statistical methods’ are used as just tools here, however, they are vital part of methodology [and so should be given due importance].

COMMENTS: In my opinion the title should have been just “impact of a multisectoral program for women Empowerment” because [in ‘Methods’ section of ‘abstract’ it is clarified by saying ‘The interventions included community dialogues on the value of girls (violence prevention), a conditional cash transfer (education), weekly group meetings with health and life skills training (health), and financial literacy training and savings activities (wealth).’ which means that] “cash transfer” is just one component of women empowerment ‘program’. Therefore, the question is “Why inclusion of ‘a cash transfer’ in title?”. {‘Conclusion’ section of ‘abstract’ has no mention of “cash transfer” component of this ‘program’}. Similarly, mention of ‘early sexual debut and fertility’ in title is questionable. Moreover, please explain what exactly you intend to indicate by term ‘Mixed-Methods Randomized Trial’ in the title?

Further, I am doubtful about whether the used one can be classified as ‘a randomized trial design’ [not sure if just random allocation means you have used a randomized trial design]. I am not worried about the ‘label’, however, concerned about the ‘level of evidence’. When you say ‘Post-hoc analysis was carried out on girls who were 13-15-years-old at baseline’, do you mean that you performed ‘sub-group’ analysis on this group? Please clarify what is ‘Post-hoc analysis’. Part of the conclusion [The need to implement such interventions in early adolescence, before negative outcomes crystallize may mean that a long follow up period is necessary to be able to observe the full impact of the intervention as the girls’ transition into young adulthood] is not from this study, it may be the authors’ opinion formed from this experience. I wonder if the investigator(s) are allowed to include in conclusion lesson(s) learned from the study?

I feel that overall, though the study has potential, the ‘presentation’ is not ‘precise’ [I mean exact or to-the-point], particularly the ‘Background’ section. On page 10 (first sentence) is ‘A random sample of eligible girls was selected for the baseline survey’ which is appearing after ‘random allocation’. Is this random sample different than earlier one? How correct it is? What exactly ‘you are trying convey’? by this. Remember that this is a scientific/academic document and so all details should be clearly/correctly communicated. In the same paragraph, further you say ‘The baseline behavioral survey was conducted after public randomization and the start of the intervention.’, what do you mean by that? What do you mean by ‘the censored nature of the indicators’?

Many {such} confusing statements [example: For girls 13 years and older at baseline, we modelled HSV-2 prevalence in 2019 and incidence between 2017 and 2019 for the sample of girls testing negative in 2017] are found throughout the article (them may be correct, however, confusing for readers). Re-drafting of the complete manuscript is necessary, in my opinion. Information given regarding ‘Sample Size Calculations’ is not very clear or convincing. I guess, the Adolescent Girls Initiative-Kenya (AGI-K) is a different study (not made clear anywhere), if so, why the power of this study is quoted from that study protocol (reference 31, The quantitative study was powered to detect differences in the prevalence of first birth and number of grades attained between the V-only and each of the other three arms at endline, four years after the start of the intervention when girls in the sample would be 15–19 years old)? From the statement made that ‘The objective of this study is to assesses the program impacts on the primary outcome of delayed childbearing, as well as on a range of secondary outcomes, two years after its completion, when the girls were 15–19 years old’, it seems that this one is a program evaluation study [because if different than (AGI-K) study, will have a separate ‘power’]. Is not it the investigator’s responsibility to make that clear? Sample size per se of this study is alright (large enough), but the argument is not convincing.

Why ‘Power analysis’ was conducted for a two-sample proportions test when the study had four arms? Please refer to ‘Randomization’ section. What do mean while saying ‘High population density in urban Kibera meant it was possible to reach a large number of girls with excludable interventions, making an individual-level randomized design feasible’? Please make clear. Clarify whether AGI-K is the program name or is it (an independent) study?

In my considered opinion, there is no point in identifying / enumerating / highlighting such loopholes / confusing statements endlessly. I recommend complete redrafting of the manuscript.

Reviewer #2: Thank you for the opportunity to read and review this important research. The authors presented the results and findings of a mixed methods original research study that assessed impacts of the Adolescent Girls Initiative-Kenya (AGI-K) program on childbearing and associated outcomes, including prevalence or incidence of sexual intercourse and pregnancy. Overall, I believe that the study findings and results are interesting and informative of the effectiveness of interventions with violence prevention, education, health, and wealth creation components for adolescent girls in Kibera to pursue education and delay sexual debut. I suggest that the authors speak to the potential sexual and relational harms of focusing on delaying sexual debut and if they could also make recommendations for future research.

Background: The authors presented a relevant review of the literature and a good rationale for conducting this study. I suggest that the authors elaborate on the following: (1) why is adolescence a particularly vulnerable period for girls? (2) what are other risk factors for acquiring HSV-2 and are there other factors that compound an HSV-2-positive individual’s risk of HIV infection? (3) how are you defining or operationalizing “cash transfer” for this study? (4) how are you differentiating, if at all, transactional sex from survival sex, and what are the familial implications of this? Further, I suggest that the authors remain consistent in their use of “unintentional pregnancy”; for example, on page 6 where they refer to the cascading ramifications of pregnancy.

Methods: The authors presented the intervention context, theory, and quantitative and qualitative methods for this study. The authors have framed this as a mixed methods study; however, they have not included any information about their mixed methods study design which would inform the reader of the sequence of procedures and methods for integration. Further, I wondered why the authors focused on delayed sexual debut and questioned if focusing on sexual education and access to reproductive resources, for example, would not be more important; especially given the age of the participants, who are at an age where it is developmentally appropriate to begin sexual exploration.

Under interventions (page 8), I suggest that the authors provide further information about the process for fidelity assessment. Who was assessing fidelity and how? Was there a measure or checklist developed for this purpose?

Under randomization (page 9), further clarity regarding assignment to study arms is needed. As it is written, there is concern about allocation concealment and potentially participant and personnel blinding. I also suggest that the authors speak to which ethical considerations that were taken regarding the public assignment.

Under quantitative methodology – outcomes (page 10), I suggest that the authors elaborate on the meaning of the “censored nature” of indicators. Further, I suggest that the authors speak to why abortion (spontaneous or induced) was not measured. The authors otherwise provided an in-depth account of the quantitative methods for this study.

Under qualitative methodology (page 12), the authors mentioned that “transcripts were coded for common themes”. This is not a sufficient account of qualitative methods. I suggest that the authors further elaborate on their data collection and analysis processes.

Results: The results of this study are contextually interesting and relevant to the problem statement described. Given that the authors coded for common themes, I suggest that the authors either incorporate sub-headings identifying these themes or a table of themes found in the findings. Further, the authors should provide an integrated analysis of the quantitative results and qualitative findings throughout the results section to justify this as a mixed methods study.

Discussion: Overall, these authors presented that the participants who received all four components of the study intervention had better outcomes and that intervention effects increased with age. Given that statistical reporting was not possible for contraceptive use due to a small sub-sample, I urge the authors to comment on the potential mechanisms of this; especially given that their qualitative findings demonstrated that participants were aware of different contraceptives and where to get them. It would also suggest that the authors speak more about the theory that they engaged with in this study and how it informed their analysis and interpretation of the findings. Finally, it was particularly confusing on page 21 where the authors refer to this study as a longitudinal design for the first time. If it was a longitudinal design, the authors need to mention this in their methods section and provide further information about the different timepoints.

Reviewer #3: This study examines the long-term impact of a multi-sectoral programme on early sexual debut and fertility in an urban informal settlement in Kenya.

The paper offers interesting and useful contributions on the effects of combined and multi-sectoral interventions to address adolescent sexual and reproductive health issues. There are, however, a few minor issues that need to be addressed to improve the paper.

There is need for details on how the qualitative analysis was conducted, who conducted.

Although included parents, teachers, mentors and gate keepers were included in the qualitative sample, there are no results from these populations.

The authors might consider using more simplified statistical language for readers. For example, a marginal reduction of 0.09 SD.

There are a few grammatical checks that need to be done e.g. check for repetition page 7, intervention context.

6. PLOS authors have the option to publish the peer review history of their article (what does this mean?). If published, this will include your full peer review and any attached files.

Reviewer #1: No

Reviewer #2: **Yes: **Rebecca Balasa

Reviewer #3: **Yes: **Joyce Wamoyi

---

## [Author Response · Author response to Decision Letter 0]

14 Sep 2021

Response: We have reviewed the PLOS ONE formatting sample and have aligned the manuscript accordingly. 

2. Please include the trial registration number in the Abstract. 

Response: The trial registration number has been included in the abstract and methodology section. 

3. Thank you for submitting your clinical trial to PLOS ONE and for providing the name of the registry and the registration number. The information in the registry entry suggests that your trial was registered after patient recruitment began. PLOS ONE strongly encourages authors to register all trials before recruiting the first participant in a study.

1) your reasons for your delay in registering this study (after enrolment of participants started);

Response: This has now been addressed:

The study was designed and our proposal (including primary and secondary outcomes, intervention arms, randomization design and questionnaires) submitted to and approved by 1) the Population Council Institutional Review Board (p661) and 2) the Kenyan AMREF Ethics and Scientific Review Committee (p143-2014) in 2014. In addition, research permits were obtained from the Kenyan National Commission for Science Technology and Innovation (P/18/6952/25330). Consequently, we followed all procedures required by the supporting institutions (including donors) and country where the work was implemented and the study had all required ethical approvals prior to contact with any subjects or enrollment, which began in 2015. 

At the time of our approved IRB submissions in 2014, however, we were unfamiliar with the process and importance of also formally registering a non-medical trial like this elsewhere, e.g., at ISRCTN. Following the year-long effort required to initiate the study in the two sites, in late 2015 we drafted a more formal study protocol (Austrian et al. 2016) and in the process of submitting that article to BMC Public Health became aware of the need to formally register the trial for journal submission —and did so registering the unchanged trial there as well. We now make this clearer in the manuscript, under ethical approval and funding. 

References:

Austrian, K., E. Muthengi, J. Mumah, E. Soler-Hampejsek, C. Kabiru, B. Abuya and J.A. Maluccio. 2016. The Adolescent Girls Initiative-Kenya (AGI-K): Study Protocol. BMC Public Health, 2016, 16:210. doi

Trial registry (assigned December 24, 2015 as part of submission process for Austrian et al. 2016). ISRCTN registry: ISRCTN77455458. https://doi.org/10.1186/ISRCTN77455458

2) Confirmation that all related trials are registered by stating: “The authors confirm that all ongoing and related trials for this drug/intervention are registered”.

Response: The phrase has been included in the ethical approval and funding section of the manuscript. 

4. Please include your tables as part of your main manuscript and remove the individual files. Please note that supplementary tables (should remain/be uploaded) as separate "supporting information" files

Response: The tables have now been included as part of the manuscript and individual table files have been removed. We have also now listed the supplementary tables under “supporting information” and have uploaded each table as a separate file. 

5. Please include additional information regarding the survey or questionnaire used in the study and ensure that you have provided sufficient details that others could replicate the analyses. For instance, if you developed a questionnaire as part of this study and it is not under a copyright more restrictive than CC-BY, please include a copy, in both the original language and English, as Supporting Information. In addition, please provide the interview guides used in the qualitative study

Response: This has now been addressed. We have included the links to publicly available questionnaires and interview guides under “Supporting information”. 

Link to publicly available questionnaire: https://www.popcouncil.org/uploads/pdfs/2021SBSR_AGI-K_EndlineSurveyInstruments.xlsx

Link to publicly available interview guides: https://www.popcouncil.org/uploads/pdfs/2021SBSR_AGI-K_MidlineQualInterviewGuides.pdf

6. Thank you for stating the following in the Competing Interests section:

The authors have declared that no competing interests exist.

We note that one or more of the authors are employed by a commercial company: Independent Consultant

6a. Please provide an amended Funding Statement declaring this commercial affiliation, as well as a statement regarding the Role of Funders in your study. If the funding organization did not play a role in the study design, data collection and analysis, decision to publish, or preparation of the manuscript and only provided financial support in the form of authors' salaries and/or research materials, please review your statements relating to the author contributions, and ensure you have specifically and accurately indicated the role(s) that these authors had in your study. You can update author roles in the Author Contributions section of the online submission form.

Response: Erica Soler-Hampejsek is a self-employed independent research consultant and not associated with any commercial company. Therefore, we declare there is no commercial affiliation. 

Under Financial disclosure we now indicate: “The funder approved the trial design and provided support in the form of salaries for authors [BK, KA, ESH, NM, YDW, BA, EM, FM, JK and JAM] but did not have any additional role in study design, data collection and analysis, preparation of the manuscript or decision to publish.” We have reviewed the statements relating to the author contributions and confirm that they accurately reflect the roles of the authors in the study. 

6b. Please also provide an updated Competing Interests Statement declaring this commercial affiliation along with any other relevant declarations relating to employment, consultancy, patents, products in development, or marketed products, etc. 

Response: As noted above, there is no commercial affiliation. We believe, therefore, that the revised funding and competing interest statements as well as information included in the cover letter are accurate.

Response: This has now been addressed. 

8. We note that Figure 2 in your submission contain copyrighted images. All PLOS content is published under the Creative Commons Attribution License (CC BY 4.0), which means that the manuscript, images, and Supporting Information files will be freely available online, and any third party is permitted to access, download, copy, distribute, and use these materials in any way, even commercially, with proper attribution. For more information, see our copyright guidelines: http://journals.plos.org/plosone/s/licenses-and-copyright.

8a. You may seek permission from the original copyright holder of Figure(s) [#] to publish the content specifically under the CC BY 4.0 license.

Response: In our submission we indicated the figure was reproduced; more accurately, for the submitted manuscript the figure in question is a modification of Fig 1 “AGI-K Theory of Change” from Austrian et al. (2016) Open Access, available at https://bmcpublichealth.biomedcentral.com/track/pdf/10.1186/s12889-016-2888-1.pdf. Because that article is published under open access all material in it is freely reproducible as indicated on its first page. The “article is distributed under the terms of the Creative Commons Attribution 4.0 International License (https://creativecommons.org/licenses/by/4.0/), which permits unrestricted use, distribution, and reproduction in any medium, provided you give appropriate credit to the original author(s) and source, provide a link to the Creative Commons license, and indicate if changes were made.”

Reference:

Austrian K, Muthengi E, Mumah J, Soler-Hampejsek E, Kabiru C, Abuya B, et al. The adolescent girls initiative-Kenya (AGI-K): study protocol. BMC public health. 2016;16(1):210. doi

8b. If you are unable to obtain permission from the original copyright holder to publish these figures under the CC BY 4.0 license or if the copyright holder’s requirements are incompatible with the CC BY 4.0 license, please either i) remove the figure or ii) supply a replacement figure that complies with the CC BY 4.0 license. Please check copyright information on all replacement figures and update the figure caption with source information. If applicable, please specify in the figure caption text when a figure is similar but not identical to the original image and is therefore for illustrative purposes only.

Response: Based on our comments above, this is now not applicable. Please see authors response to 8a. 

Additional Editor Comments:

Per PLOS ONE guidelines, please explain in the methods why the trial was not prospectively registered. 

Response: This has now been addressed (please see response to comment #1 above for further details). All ethical requirements for the research were completed in 2014, prior to contacting or enrolling participants. As requested, the methods section (under “Ethical Approval and Funding”) now includes the following: 

“Upon completion of a draft study protocol paper in 2015, the unchanged trial was also retrospectively registered in the ISCRTN registry (ISRCTN77455458) as was required for journal submission; the authors confirm that all ongoing and related trials for this intervention are registered.”

Reviewers' comments:

Reviewer's Responses to Questions

Comments to the Author

1. Is the manuscript technically sound, and do the data support the conclusions?

Reviewer #1: Partly

Reviewer #2: Partly

Reviewer #3: Yes

Response: We have revised the manuscript addressing all related comments by the reviewers. Please see below.________________________________________

2. Has the statistical analysis been performed appropriately and rigorously? 

Reviewer #1: I Don't Know

Reviewer #2: Yes

Reviewer #3: Yes

Response: We have clarified further the statistical analyses in response to reviewer #1 comments. Please see below. ________________________________________

3. Have the authors made all data underlying the findings in their manuscript fully available?

Reviewer #1: Yes

Reviewer #2: Yes

Reviewer #3: No

Response: We have now provided under “supporting information” links to publicly available questionnaires, interviewer guides and data.

4. Is the manuscript presented in an intelligible fashion and written in standard English?

Reviewer #1: No

Reviewer #2: No

Reviewer #3: Yes

Response: We have revised the manuscript addressing all related comments by the reviewers. Please see below.________________________________________

5. Review Comments to the Author

Reviewer #1: 

1. Important note: This review pertains only to ‘statistical aspects’ of the study and so ‘clinical aspects’ [like medical importance, relevance of the study, ‘clinical significance and implication(s)’ of the whole study, etc.] are to be evaluated [should be assessed] separately/independently. Further please note that any ‘statistical review’ is generally done under the assumption that (such) study specific methodological [as well as execution] issues are perfectly taken care of by the investigator(s). This review is not an exception to that and so does not cover clinical aspects {however, seldom comments are made only if those issues are intimately / scientifically related & intermingle with ‘statistical aspects’ of the study}. Agreed that ‘statistical methods’ are used as just tools here, however, they are vital part of methodology [and so should be given due importance].

Response: Thank you for the careful attention to the statistical aspects of our submission. We are in complete agreement that statistical tools used are a vital part of the methodology and have reworked the text to make them clearer. Below we indicate how we have addressed and incorporated your comments in the substantially revised manuscript which also incorporates changes in response to the other reviewers who as you describe it focused more on the relevance of the study, i.e., clinical significance and implications.

2. COMMENTS: In my opinion the title should have been just “impact of a multisectoral program for women Empowerment” because [in ‘Methods’ section of ‘abstract’ it is clarified by saying ‘The interventions included community dialogues on the value of girls (violence prevention), a conditional cash transfer (education), weekly group meetings with health and life skills training (health), and financial literacy training and savings activities (wealth).’ which means that] “cash transfer” is just one component of women empowerment ‘program’. Therefore, the question is “Why inclusion of ‘a cash transfer’ in title?”. {‘Conclusion’ section of ‘abstract’ has no mention of “cash transfer” component of this ‘program’}. Similarly, mention of ‘early sexual debut and fertility’ in title is questionable. Moreover, please explain what exactly you intend to indicate by term ‘Mixed-Methods Randomized Trial’ in the title?

Response: We have modified the title to be: 

Impacts of multisectoral cash plus programs after four years in an urban informal settlement: Adolescent Girls Initiative-Kenya (AGI-K) randomized trial

As suggested, we have modified the title to better reflect the complex nature of the interventions and have removed the “mixed-methods” designation. We continue to signal that the interventions include a cash transfer component to align with research on related programs and make the article easily accessible to those carrying out bibliographic searches for research on programs with cash transfers, particularly as these have become even more prevalent during the pandemic. As recommended, because the paper examines several important outcomes, we dropped direct mention in the title of only two of them (“early sexual debut and fertility”). Reviewer #2 also questioned whether it was appropriate to frame the study as having a “mixed-methods approach,” so we have removed that designation from the title and text. Nevertheless, we believe the qualitative research remains an important component of the evidence we provide. Consequently, we continue to report the relevant qualitative findings (which we believe complement and strengthen the statistical quantitative results) and in response to the other reviewers now more carefully describe the qualitative research design that accompanied the quantitative randomized trial. 

3. Further, I am doubtful about whether the used one can be classified as ‘a randomized trial design’ [not sure if just random allocation means you have used a randomized trial design]. I am not worried about the ‘label’, however, concerned about the ‘level of evidence’. When you say ‘Post-hoc analysis was carried out on girls who were 13-15-years-old at baseline’, do you mean that you performed ‘sub-group’ analysis on this group? Please clarify what is ‘Post-hoc analysis’. Part of the conclusion [The need to implement such interventions in early adolescence, before negative outcomes crystallize may mean that a long follow up period is necessary to be able to observe the full impact of the intervention as the girls’ transition into young adulthood] is not from this study, it may be the authors’ opinion formed from this experience. I wonder if the investigator(s) are allowed to include in conclusion lesson(s) learned from the study?

Response: The AGI-K program randomly (i.e., experimentally) allocated one of four packages of interventions at the individual level to a sample of eligible girls residing in Kibera, Kenya. The study protocol was submitted to the Institutional Review Board at the Population Council and approved in 2014. We argue, therefore, that it has a randomized trial design consistent with common academic use of that concept, for example in public health and economics. Moreover, we argue that results from statistical analyses of the randomized trial provide rigorous evidence of the causal effect of each of the three packages of interventions (or study arms: VE, VEH, VEHW) compared with the V-only intervention study arm. In addition, the design permits estimation of the causal effects for three other possible comparisons (i.e., VEH vs VE, VEHW vs VE and VEHW vs VEH). (For completeness, we note in this response that we intentionally do not refer to it as a randomized controlled trial since as elaborated on in the paper it was not feasible to include a pure control group.) 

Regarding “post-hoc analysis”, yes that is correct, the analysis of girls 13+ years old at baseline is a subgroup analysis. We referred to it as ‘post-hoc’ because it had not been pre-specified or outlined in the published protocol paper (Austrian et al. 2016). We now clarify that meaning and the logic for examining this subgroup, which was that 1) by virtue of being older at endline they were more likely to have begun having sex, become pregnant or given birth by the time of the survey and 2) they were the subgroup for which the additional HSV-2 measurements were taken. 

Finally, we have modified the conclusions, including the highlighted statement [“The need to implement such interventions in early adolescence…”]. The current study provides evidence that the AGI-K packages of interventions affected mediating factors and delayed sexual debut for some study arms and groups. This points to the possibility that these types of interventions may be a promising approach for reducing early fertility. Even at endline in 2019, however, many girls were still only in their relatively young teens. Therefore, measurement of the cohort at older ages is needed for a full assessment of the effects of intervening in early adolescence on fertility-related outcomes for young adult females. Thus, longer-term follow-up is a priority for future research, something reviewer #2 asked us to comment on. Presently, we are seeking funding to do such follow-up.

More specifically, we conclude in the discussion section the following: 

“Future research should be carried out on understanding the impact of short-term cash-plus multisectoral interventions implemented in early adolescence on outcomes that become increasingly common in later adolescence and early adulthood, such as secondary school completion, income generation and contraceptive use.”

Reference:

Austrian K, Muthengi E, Mumah J, Soler-Hampejsek E, Kabiru C, Abuya B, et al. The adolescent girls initiative-Kenya (AGI-K): study protocol. BMC public health. 2016;16(1):210. doi

4. I feel that overall, though the study has potential, the ‘presentation’ is not ‘precise’ [I mean exact or to-the-point], particularly the ‘Background’ section. On page 10 (first sentence) is ‘A random sample of eligible girls was selected for the baseline survey’ which is appearing after ‘random allocation’. Is this random sample different than earlier one? How correct it is? What exactly ‘you are trying convey’? by this. Remember that this is a scientific/academic document and so all details should be clearly/correctly communicated. In the same paragraph, further you say ‘The baseline behavioral survey was conducted after public randomization and the start of the intervention.’, what do you mean by that? What do you mean by ‘the censored nature of the indicators’?

Response: (a) We have clarified the text including removing the confusing reference to random allocation to study arm alongside sampling for the survey, which are distinct. (b) We removed the sentence beginning with “The baseline behavioral survey was conducted…” and the timing of the quantitative baseline survey is now described in the “Randomization and Data Collection” section. (c) Last, we removed the term “censored nature” which was indeed uninformative (and confusing to Reviewer #2 as well), rewriting the description of the primary outcome variables as follows:

The primary outcomes include binary 0/1 variables measured at endline and equal to one if the girl had ever: 1) had sex; 2) been pregnant; or 3) given birth.

Here we provide further explanation of (a): Because the order in which the different aspects of the study occurred and the process is crucial, we summarize them here (and have rewritten the text accordingly in “Randomization and Data Collection”). First, because there was no recent census available in the study area, we implemented a household census to identify the set of all girls living in the study area that were potentially eligible for the program. We then used this set to generate a list with only one eligible girl per household to carry out random allocation to study arms. In households with one eligible girl the girl was selected for the list. In households with more than one eligible girl, one girl was randomly selected for the list. Effectively this means allocation to study arm was at the household level since when the program began all other eligible girls in the household were invited to participate in the same study arm as the girl selected for the list.

This process yielded a list of 3,296 girls (each representing a single household) potentially eligible for the program after the initial household listing (See Figure 4); reiterating, this included exactly one girl per household. Using anonymous identification numbers we generated, these girls were then each randomly allocated to one of the four study arms in a public lottery. At this stage, however, no girls or households were contacted yet regarding the treatment status which remained concealed.

Subsequently a quantitative baseline survey was administered, prior to unblinding of study arm assignment to girls (and their households) and to the start of the program. Baseline enumerators were similarly blinded to the girl’s study arm assignment. The baseline survey targeted all girls on the list randomized to study arms, but reconfirmed eligibility prior to carrying out an interview. At a later date study arm assignment was revealed to the interviewed girl and all other eligible girls in her household, if any, were invited to participate in the program in the same study arm. 

The baseline survey was conducted February–April 2015. All girls interviewed at baseline were targeted for longitudinal follow-up two years later at the end of the program (May–July 2017) and then four years later at endline (April–July 2019) (Fig 4). HSV-2 was collected for older girls (13 years old or older at baseline) in 2017 and 2019, when they were 15 years old and above.

5. Many {such} confusing statements [example: For girls 13 years and older at baseline, we modelled HSV-2 prevalence in 2019 and incidence between 2017 and 2019 for the sample of girls testing negative in 2017] are found throughout the article (them may be correct, however, confusing for readers). Re-drafting of the complete manuscript is necessary, in my opinion. Information given regarding ‘Sample Size Calculations’ is not very clear or convincing. I guess, the Adolescent Girls Initiative-Kenya (AGI-K) is a different study (not made clear anywhere), if so, why the power of this study is quoted from that study protocol (reference 31, The quantitative study was powered to detect differences in the prevalence of first birth and number of grades attained between the V-only and each of the other three arms at endline, four years after the start of the intervention when girls in the sample would be 15–19 years old)? From the statement made that ‘The objective of this study is to assesses the program impacts on the primary outcome of delayed childbearing, as well as on a range of secondary outcomes, two years after its completion, when the girls were 15–19 years old’, it seems that this one is a program evaluation study [because if different than (AGI-K) study, will have a separate ‘power’]. Is not it the investigator’s responsibility to make that clear? Sample size per se of this study is alright (large enough), but the argument is not convincing.

Response: We apologize for portions of the text that were confusing and have redrafted the manuscript for clarity, including expanding in places as necessary. Here we respond directly to your various points.

First, the paper directly analyzes the AGI-K study in Kibera, something we now clarify starting from including AGI-K in the title of the paper to avoid confusion. The AGI-K study was carried out in two distinct sites that are analyzed separately (Kibera and Wajir). AGI-K program evaluation results from Kibera are presented in the present paper and results from the other study site (Wajir) reported elsewhere (Austrian et al. 2021). Therefore, the power calculations discussed are exactly the ones done for this study. (Please see response to your comment #6 for more detail on the power analyses.)

AGI-K comprised intervention packages targeting girls 11–14 years old at baseline in 2015 and lasting for two years, i.e., from 2015 to 2017. The longitudinal endline survey was planned for, and carried out, in 2019. This was four years after the start of the AGI-K interventions (in 2015) and two years after their end (in 2017). Girls who were 11–14 years old at baseline in 2015 would be 15–18 years old at endline in 2019. Therefore, power calculations were based on outcomes for girls 15–18 at endline. 

Please see the revised “Sample Size and Power Analysis” section where we have clarified the approach taken which was an assessment of minimum detectable effects (MDE) based on potential available sample size. 

Regarding our prior statement: “For girls 13 years and older at baseline, we modelled HSV-2 prevalence in 2019….” We have modified the text in the “Outcomes” subsection as follows:

“We also directly examined HSV-2, an important health outcome in its own right and also valuable as an objectively measured outcome that can be used to corroborate self-reported sexual activity. Trained HIV services counsellors collected biological blood specimens via finger prick for girls 15 years old and older in 2017 and at endline in 2019 that were tested for HSV-2 (S8 Text). For the subgroup of girls with HSV-2 measurements we examined binary 0/1 variables equal to one if the girl: 1) tested positive for HSV-2 at endline (i.e., prevalence in 2019); and 2) tested positive for HSV-2 at endline having tested negative in 2017 at the two-year follow-up, indicating individuals seroconverting from negative to positive between 2017 and 2019 (i.e., incidence between 2017 and 2019).”

Reference:

Austrian K, Soler-Hampejsek E, Kangwana B, Maddox N, Diaw M, Wado Y.D, Abuya B, Muluve E, Mbushi F, Mohammed H, Aden A and Maluccio JA. Impacts of multisectoral cash plus programs on marriage and fertility after four years in pastoralist Kenya: a randomized trial. Submitted manuscript, 2021.

6. (a) Why ‘Power analysis’ was conducted for a two-sample proportions test when the study had four arms? Please refer to ‘Randomization’ section. (b) What do mean while saying ‘High population density in urban Kibera meant it was possible to reach a large number of girls with excludable interventions, making an individual-level randomized design feasible’? Please make clear. (c) Clarify whether AGI-K is the program name or is it (an independent) study?

Response: 

(a) We now clarify that all the principal hypotheses outlined in the analysis plan for AGI-K in the study protocol are comparisons across two study arms at a time. Therefore, we argue the appropriate power calculations for minimum detectable effects are the two-sample tests we used. 

(b) We have also rewritten/clarified the ideas underlying the statement beginning “High population density in urban Kibera….”. The text in the “Randomization and Data Collection” section now reads: 

High population density and widespread availability of schools in Kibera made it possible to reach a large number of girls there with different intervention packages. For example, it was feasible to offer VE to one girl in the study area and VEH to another, inviting the second girl to participate in the health intervention girls meetings while the first was excluded from that intervention component. Therefore, we implemented an individual-level randomized design in which the unit of randomization was the girl (and her household).

(c) Regarding your last question, yes, the Adolescent Girls Initiative-Kenya (AGI-K) is the name of the program being analyzed in this paper (and not an independent or different study). We now clarify this from the outset including directly naming the program in the revised title (see our responses to your comments #2 and #5 above). 

In my considered opinion, there is no point in identifying / enumerating / highlighting such loopholes / confusing statements endlessly. I recommend complete redrafting of the manuscript.

Response: Thank you for your valuable comments. We have now redrafted the entire manuscript as requested, highlighting major modifications in red and important additions to the manuscript in response to reviewer comments in green. 

Reviewer #2: 

1. Thank you for the opportunity to read and review this important research. The authors presented the results and findings of a mixed methods original research study that assessed impacts of the Adolescent Girls Initiative-Kenya (AGI-K) program on childbearing and associated outcomes, including prevalence or incidence of sexual intercourse and pregnancy. Overall, I believe that the study findings and results are interesting and informative of the effectiveness of interventions with violence prevention, education, health, and wealth creation components for adolescent girls in Kibera to pursue education and delay sexual debut. I suggest that the authors speak to (a) the potential sexual and relational harms of focusing on delaying sexual debut and (b) if they could also make recommendations for future research.

Response: Thank you very much for the detailed review and valuable comments allowing us to clarify confusing aspects of the manuscript and pushing us to think more carefully about our interpretations and the implications of the research. Below we respond to your comments and indicate how we have revised the manuscript to incorporate them.

(a) As we now make clearer in the theory of change, the program aimed to delay fertility in Kibera by delaying early sexual debut (intercourse) and through improved knowledges and practice of contraception and family planning, not solely through delayed sexual debut. At endline, all subjects were 18 years old or younger. 

(b) 

We have now added in recommendations for future research in the discussion section as follows:

“Future research should be carried out on understanding the impact of short-term cash-plus multisectoral interventions implemented in early adolescence on outcomes that become increasingly common in later adolescence and early adulthood, such as secondary school completion, income generation and contraceptive use.”

2. Background: The authors presented a relevant review of the literature and a good rationale for conducting this study. I suggest that the authors elaborate on the following: (1) why is adolescence a particularly vulnerable period for girls? (2) what are other risk factors for acquiring HSV-2 and are there other factors that compound an HSV-2-positive individual’s risk of HIV infection? (3) how are you defining or operationalizing “cash transfer” for this study? (4) how are you differentiating, if at all, transactional sex from survival sex, and what are the familial implications of this? (5) Further, I suggest that the authors remain consistent in their use of “unintentional pregnancy”; for example, on page 6 where they refer to the cascading ramifications of pregnancy.

Response: 

(1) In response to your first question: The main reasons adolescence is a particularly vulnerable period is that girls undergo rapid change during those years as manifested by physical, cognitive, social, emotional and sexual development. We now explain this in the introduction and include additional relevant references. 

References:

Patton GC, Sawyer SM, Santelli JS, Ross DA, Afifi R, Allen NB, Arora M, Azzopardi P, Baldwin W, Bonell C, Kakuma R, Kennedy E, Mahon J, McGovern T, Mokdad AH, Patel V, Petroni S, Reavley N, Taiwo K, Waldfogel J, Wickremarathne D, Barroso C, Bhutta Z, Fatusi AO, Mattoo A, Diers J, Fang J, Ferguson J, Ssewamala F, Viner RM. Our future: a Lancet commission on adolescent health and wellbeing. Lancet. 2016 Jun 11;387(10036):2423-78. doi: 10.1016/S0140-6736(16)00579-1. Epub 2016 May 9. PMID: 27174304; PMCID: PMC5832967.

Suleiman AB, Dahl RE. Leveraging Neuroscience to Inform Adolescent Health: The Need for an Innovative Transdisciplinary Developmental Science of Adolescence. J Adolesc Health. 2017 Mar;60(3):240-248. doi: 10.1016/j.jadohealth.2016.12.010. PMID: 28235453.

Global Accelerated Action for the Health of Adolescents (AA-HA!): guidance to support country implementation. Summary. Geneva: World Health Organization; 2017 (WHO/FWC/MCA/17.05). Licence: CC BY-NC-SA 3.0 IGO.

(2) In response to your second question: Other known risk factors for HSV-2 infection include having multiple sexual partners, having a previous history of sexually transmitted infections (STIs), and low education or socioeconomic status (Odhiambo et al., 2017; Mugo et al., 2011). Similar to HIV, the main method of HSV-2 transmission is through sexual contact, however (Johnston et al., 2016), in rare instances HSV-2 virus can be transmitted to neonates during delivery, usually resulting in neonatal death or severe disability (Corey et al., 2009; Brown et al., 1997) 

In addition, HSV-2 has been shown to increase an individual’s susceptibility to HIV infection by two to three fold and transmission of HIV infection by up to five-fold (Odhiambo et al., 2017; Mugo et al., 2011) mainly through high concentrations of activated CD4 positive T cells in the genital area which are targeted by HIV and increased likelihood of breakage of the mucosal layer caused by these cells that creates an entry point for the HIV virus, both in the asymptomatic and symptomatic phase (Van de Perre et al., 2008; Gutierrez et al., 2007). Susceptibility is heightened during the symptomatic phase, through the presence of genital ulcers which are more prevalent during the first few years of infection (Van de Perre et al., 2008; Naswa et al., 2010). 

We now indicate the additional risk factors for HSV-2 in the introduction and provide additional references for its links to HIV, but do not explicitly incorporate the details on risk for HIV which we do not directly study.

References: 

Akinyi B, Odhiambo C, Otieno F, Inzaule S, Oswago S, Kerubo E, et al. Prevalence, incidence and correlates of HSV-2 infection in an HIV incidence adolescent and adult cohort study in western Kenya. PloS one. 2017;12(6):e0178907.

Brown ZA, Selke S, Zeh J, Kopelman J, Maslow A, Ashley RL, et al. The acquisition of herpes simplex virus during pregnancy. The New England journal of medicine. 1997;337(8):509-15.

Corey L, Wald A. Maternal and neonatal herpes simplex virus infections. The New England journal of medicine. 2009;361(14):1376-85.

Johnston C, Corey L. Current Concepts for Genital Herpes Simplex Virus Infection: Diagnostics and Pathogenesis of Genital Tract Shedding. Clinical microbiology reviews. 2016;29(1):149-61.

Mugo N, Dadabhai SS, Bunnell R, Williamson J, Bennett E, Baya I, et al. Prevalence of herpes simplex virus type 2 infection, human immunodeficiency virus/herpes simplex virus type 2 coinfection, and associated risk factors in a national, population-based survey in Kenya. Sexually transmitted diseases. 2011;38(11):1059-66.

Naswa S, Marfatia YS. Adolescent HIV/AIDS: Issues and challenges. Indian journal of sexually transmitted diseases and AIDS. 2010;31(1):1-10.

Van de Perre P, Segondy M, Foulongne V, Ouedraogo A, Konate I, Huraux JM, et al. Herpes simplex virus and HIV-1: deciphering viral synergy. The Lancet Infectious diseases. 2008;8(8):490-7 

(3) In response to question three: The AGI-K program provided what are usually referred to as conditional cash transfers. These are transfers made upon completion of a specific condition, in contrast to unconditional or “no-strings-attached” cash transfers which have no conditions or co-responsibilities on the part of the beneficiary household. The transfers and related conditions are described in Figure 2. For example, the first transfer to the household per school term was made upon verification that the girl had enrolled in school for the term. In providing the conditional cash transfers, AGI-K emphasized that funds were meant to support the girl in her schooling, but actual use of the money was not monitored. Logistically, the transfers were made via electronic payments to the personal bank account of the beneficiary household. Because not all beneficiaries had bank accounts at the outset of the program, AGI-K assisted beneficiaries in opening them. 

(4) In response to question four: We assessed transactional sex mainly from the qualitative data. The question posed to girls as well as other stakeholders regarding sexual behavior was “can you describe the different types of romantic/ sexual relationships that girls may have and the motivations for that relationship?” The responses indicated that motivations for girls engaging in sex included money, gifts and peer and media influence. We define transactional sex to mean having sex in anticipation of or actual exchange for money or gifts but are unable to go further and differentiate between transactional sex and survival sex for this young cohort. We did not analyze transactional sex using the quantitative data.

The qualitative data suggest that some parents are aware of their daughters engaging in sexual relationships in exchange for money and gifts but do not address it because they benefit indirectly from the receipt of the money or gifts. However, it is not clear from the data how common this practice might be. 

(5) In response to question five: Our paper examines “early pregnancy” without trying to separate out or directly examine “unintended” pregnancies. Hence, we have corrected our inconsistent and confusing use of the concept of “unintended pregnancy” by removing reference to it when discussing our hypotheses and analyses but continue to use it when referring to other literature that does directly address it. 

We note that one reason for not directly analyzing unintended pregnancies is that it is difficult to reliably gather information about them. The survey did ask those who were currently pregnant “At the time you became pregnant, did you want to become pregnant then, did you want to wait until later, or did you not want the pregnancy at all?” In 2019 for N=43 responding, 28% indicated they had wanted the pregnancy then, 30% indicated later and 42% indicated not at all. 

3. Methods: The authors presented the intervention context, theory, and quantitative and qualitative methods for this study. (a) The authors have framed this as a mixed methods study; however, they have not included any information about their mixed methods study design which would inform the reader of the sequence of procedures and methods for integration. (b) Further, I wondered why the authors focused on delayed sexual debut and questioned if focusing on sexual education and access to reproductive resources, for example, would not be more important; especially given the age of the participants, who are at an age where it is developmentally appropriate to begin sexual exploration.

Response: 

(a) Reviewer #1 also queried whether it was appropriate to frame the study as having a “mixed-methods approach,” so we have removed that designation from the title and text. Nevertheless, we believe the qualitative research remains an important component of the overall evidence we provide. Consequently, we continue to report the relevant qualitative findings (which we believe complement and strengthen the statistical quantitative results) and in response to your comments and those of Reviewer #3 now more fully describe the qualitative research design that accompanied the quantitative randomized trial. 

(b) Based on our theory of change we hypothesize that the education on health and life skills received through the girls’ groups will increase knowledge on sexual and reproductive health (SRH) which will in turn result in delayed sexual debut and increased FP use. Two years into the implementation of the interventions there was a significant increase in sexual and reproductive knowledge for girls who received the health component of the intervention. This has been documented in a paper examining the effects of the program after two years (Austrian et al., 2021). Four years after the implementation of the intervention we continue to observe improved SRH knowledge as well as some evidence of delayed sexual debut. We go further to recommend future research be carried out to understand the impact of this and other similar short-term interventions on girls as they transition into adulthood, including the impact of such an intervention on use of contraception, as more girls become sexually active.

Reference:

 Austrian, K., Soler-Hampejsek E, Kangwana B, Wado Y.D, Abuya B and Maluccio J.A. Impacts of two-year multisectoral cash plus programs on young adolescent girls’ education, health and economic outcomes: Adolescent Girls Initiative-Kenya (AGI-K) randomized trial. 2021 submitted manuscript.

4. Under interventions (page 8), I suggest that the authors provide further information about the process for fidelity assessment. Who was assessing fidelity and how? Was there a measure or checklist developed for this purpose? 

Response: We have now included a description of the processes, which included an electronic monitoring database for program activities (fulfillment of conditions by girls, all transfers, details of all girls group meetings held) as well as monitoring of the mentors and their groups. Please see “Interventions” section. 

5. Under randomization (page 9), (a) further clarity regarding assignment to study arms is needed. As it is written, there is concern about allocation concealment and potentially participant and personnel blinding. (b) I also suggest that the authors speak to which ethical considerations that were taken regarding the public assignment. 

Response: (a) We have reworked the section on “Randomization and Data Collection” to improve clarity. Given the nature of the interventions it was of course not possible during program implementation for participants or NGO implementing personnel to be blinded to the study arm of an individual girl once the intervention began. The random allocation, however, was not known to enumerators or beneficiaries (i.e., it was still concealed) at the time of the baseline survey. During the two-year follow-up survey (analyzed in a separate paper, Austrian et al. 2021) there were questions about the program at the end of the survey so at that stage in the interview enumerators would become aware of the study arm based on skip patterns (for example, questions about girls group meetings were asked only of those in VEH or VEHW study arms). These questions were not asked in the endline survey, however, making it less likely enumerators would have been aware of the previously assigned study arm for the program that had ended two years earlier. 

(b) Ethical considerations of public assignment: We outline ethical approval for the randomized trial in the section “Ethical Approval and Funding.” The principal ethical justification for carrying out the randomized trial with different beneficiaries receiving different packages of interventions was to estimate the unknown marginal benefit as each single-sector intervention was added to the package relative to the V-only intervention. In this situation of equipoise randomization was justified. For community acceptance of the research intervention, it was important to do the allocation in a transparent manner with community stakeholders present, so randomization was done publicly, directed by the Population Council. First, the four interventions and study arms were described after which the procedure for randomization detailed and then carried out and formally recorded with the Population Council until beneficiaries were notified at a later date. 

Reference:

Austrian, K., E. Soler-Hampejsek, B. Kangwana, Y.D. Wado, B. Abuya and J.A. Maluccio. Impacts of two-year multisectoral cash plus programs on young adolescent girls’ education, health and economic outcomes: Adolescent Girls Initiative-Kenya (AGI-K) randomized trial. 2021 submitted manuscript. 

6. Under quantitative methodology – outcomes (page 10), (a) I suggest that the authors elaborate on the meaning of the “censored nature” of indicators. (b) Further, I suggest that the authors speak to why abortion (spontaneous or induced) was not measured. The authors otherwise provided an in-depth account of the quantitative methods for this study. 

Response: 

(a) ‘the censored nature of the indicators’: We have removed this confusing (and unnecessary) text, rewriting the description of the primary outcome variables as below: 

The primary outcome measures include binary 0/1 variables measured at endline and equal to one if the girl had ever: 1) had sex; 2) been pregnant; or 3) given birth.

(b) Although abortion in Kenya is common (see below), it is illegal in most situations and therefore a particularly sensitive topic. Therefore, we did not ask about it directly, because of concern regarding the reliability of the responses about an illegal activity and the possibility of then jeopardizing full responses and cooperation with the many other questions in the survey or even leading to refusal to participate. 

These other questions included multiple questions about current and prior pregnancy that give us confidence that measurement of pregnancy is accurate despite not asking directly about abortion. 

In particular, in addition to asking directly about prior or current pregnancies, the survey included redundancies to accurately capture prior pregnancies. For all girls reporting they had had sex, we asked “Have you ever had a pregnancy that miscarried or ended in a stillbirth?” Consequently, our measure of pregnancy does include spontaneous abortion and possibly induced abortion depending on how the girl responded if that had been her experience. We also separately asked “Sometimes a girl becomes pregnant when she does not want to be. Have you ever been pregnant when you did not want to be?” Therefore, we have high confidence in the ever-pregnant variable. 

Further background details: Abortion is common in Kenya although most of it is illegal. Estimates from 2002 indicate the abortion rate (abortions per 1000 women of childbearing age) was 46 and the abortion ratio (abortions per 100 pregnancies) was 26 (Guttmacher 2012). Induced abortion in Kenya is illegal in most situations, but is legal in cases in which the mother’s health or life is at risk since 2010. Prior to that it had been legal only to save the mother’s life. 

The 2014 Kenya Demographic and Health Survey (KDHS) does ask directly about induced abortion, including it as an addition to the question we asked: “Have you ever had a pregnancy that miscarried, was aborted, or ended in stillbirth?” Authors analysis of KDHS indicates that in the Nairobi region <3% of females aged 15–19 responded yes to this question; this was the same percent responding yes to the similar question we included in AGI-K, further bolstering confidence in our measurement of pregnancy. 

These low percentages also underscore that it would have been difficult to directly study induced abortion in this population-based survey. To measure abortion different sampling approaches are usually employed, such as sampling among women presenting for abortion services (Kabiru et al. 2016).

References:

Guttmacher Institute. Abortion and unintended pregnancy in Kenya. Series 2012, No.2 

Kabiru, C.W., B.A. Ushie, M.M. Mutua and C.O. Izugbara. Previous induced abortion among young women seeking abortion-related care in Kenya: A cross-sectional analysis. 2016. BMC Pregnancy and Childbirth 16:104. 

6. Under qualitative methodology (page 12), the authors mentioned that “transcripts were coded for common themes”. This is not a sufficient account of qualitative methods. I suggest that the authors further elaborate on their data collection and analysis processes. 

Response: Agreed. This was an important omission in our first submission also commented on by Reviewer #3. We now fully describe the data collection (in the “Randomization and Data Collection) and analysis processes (in the “Qualitative Methodology”) section. 

7. Results: The results of this study are contextually interesting and relevant to the problem statement described. Given that the authors coded for common themes, I suggest that the authors either incorporate sub-headings identifying these themes or a table of themes found in the findings. Further, the authors should provide an integrated analysis of the quantitative results and qualitative findings throughout the results section to justify this as a mixed methods study. 

Response: As recommended, we have incorporated sub-headings reflecting common themes from the qualitative results. As indicated above in our response to your Comment #3, we no longer define this as a “mixed-methods” study but instead are using findings from the qualitative data collection to help explain the quantitative findings. In the results section, therefore, we have presented the relevant qualitative findings next to the quantitative findings. We then proceed to integrate the quantitative and qualitative findings in the discussion. 

8. Discussion: Overall, these authors presented that the participants who received all four components of the study intervention had better outcomes and that intervention effects increased with age. Given that statistical reporting was not possible for contraceptive use due to a small sub-sample, I urge the authors to comment on the potential mechanisms of this; especially given that their qualitative findings demonstrated that participants were aware of different contraceptives and where to get them. It would also suggest that the authors speak more about the theory that they engaged with in this study and how it informed their analysis and interpretation of the findings. Finally, it was particularly confusing on page 21 where the authors refer to this study as a longitudinal design for the first time. If it was a longitudinal design, the authors need to mention this in their methods section and provide further information about the different timepoints. 

Response: We have now added into the discussion the potential mechanisms in which the intervention was hypothesized to have an impact on contraceptive use (paragraphs 3 and 8 of the discussion section). We also expanded the explanation of how our method of analysis and results relate to the study's proposed theory of change.

Last, we apologize for the confusion regarding the longitudinal design. We now clarify in both the abstract and the data description that this is a (prospective) longitudinal study.  

Reviewer #3: 

1. This study examines the long-term impact of a multi-sectoral programme on early sexual debut and fertility in an urban informal settlement in Kenya. The paper offers interesting and useful contributions on the effects of combined and multi-sectoral interventions to address adolescent sexual and reproductive health issues. There are, however, a few minor issues that need to be addressed to improve the paper.

Response: Thank you for the positive assessment of our paper – please see above (and in the manuscript) how we addressed your comments. 

2. There is need for details on how the qualitative analysis was conducted, who conducted. 

Response: Agreed. This was an important omission in our first submission also commented on by Reviewer #2. We now fully describe the data collection (in the “Randomization and Data Collection) and analysis processes (in the “Qualitative Methodology”) section.

3. Although included parents, teachers, mentors and gate keepers were included in the qualitative sample, there are no results from these populations. 

Response: Although the qualitative results section does not include direct quotes from every single type of respondent, responses reported by all respondents are reflected/summarized in the presentation of the results and discussion. 

4. The authors might consider using more simplified statistical language for readers. For example, a marginal reduction of 0.09 SD. 

Response: We have modified the referenced statement and others like it throughout the text. In particular, we replaced the term “marginally significant” with more direct mention of significance level (e.g., 10% significance) when appropriate. For the summary z-score measures the coefficient estimates all reflect changes in standard deviations (SD) in the summary measure so this was not altered but is now spelled out better in the quantitative methodology section. 

5. There are a few grammatical checks that need to be done e.g. check for repetition page 7, intervention context. 

Response: We have copy edited the entire manuscript, including page 7 repetition regarding the intervention context. 

6. PLOS authors have the option to publish the peer review history of their article (what does this mean?). If published, this will include your full peer review and any attached files.

Response: We are ok for the peer reviewed history of this article to be published in full. 

Do you want your identity to be public for this peer review? For information about this choice, including consent withdrawal, please see our Privacy Policy.

Reviewer #1: No

Reviewer #2: Yes: Rebecca Balasa

Reviewer #3: Yes: Joyce Wamoyi

---

## [Decision Letter · Decision Letter 1]

26 Oct 2021

PONE-D-21-06054R1Impacts of multisectoral cash plus programs after four years in an urban informal settlement: Adolescent Girls Initiative-Kenya (AGI-K) randomized trialPLOS ONE

Dear Dr. Kangwana,

Thank you for submitting your manuscript to PLOS ONE. After careful consideration, we feel that it has merit but does not fully meet PLOS ONE’s publication criteria as it currently stands. Therefore, we invite you to submit a revised version of the manuscript that addresses the points raised during the review process.

We look forward to receiving your revised manuscript.

Kind regards,

Catherine E Oldenburg

Academic Editor

PLOS ONE

Journal Requirements:

Reviewers' comments:

Reviewer's Responses to Questions

**Comments to the Author**

1. If the authors have adequately addressed your comments raised in a previous round of review and you feel that this manuscript is now acceptable for publication, you may indicate that here to bypass the “Comments to the Author” section, enter your conflict of interest statement in the “Confidential to Editor” section, and submit your "Accept" recommendation.

Reviewer #1: (No Response)

Reviewer #2: (No Response)

2. Is the manuscript technically sound, and do the data support the conclusions?

Reviewer #1: (No Response)

Reviewer #2: Yes

3. Has the statistical analysis been performed appropriately and rigorously? 

Reviewer #1: (No Response)

Reviewer #2: Yes

4. Have the authors made all data underlying the findings in their manuscript fully available?

Reviewer #1: (No Response)

Reviewer #2: (No Response)

5. Is the manuscript presented in an intelligible fashion and written in standard English?

Reviewer #1: No

Reviewer #2: No

6. Review Comments to the Author

Reviewer #1: Although there is some improvement in manuscript after revision, still there are many confusing statements and as said earlier the ‘presentation’ is not ‘precise’, to-the-point. Please check the English language [professional intervention may be needed, I guess]. I have no specific recommendation(s) and think that, ‘let the respected editor decide the future course’.

Reviewer #2: Thank you to the authors for their in-depth responses to reviewer comments and for revising this manuscript accordingly. My remaining major recommendations are as follows: (1) the authors speak to transactional sex in the background and prostitution in their findings. As it is written, the authors seem to be overlooking that children and youth cannot consent to sex work. Instead of reporting that girls in their sample engaged in prostitution, I would recommend writing that they have experience child sexual exploitation. If the context and laws in Kenya suggest otherwise, I strongly recommend that the authors highlight this; and (2) the section describing qualitative methodology and methods is still insufficient. The authors reported using grounded theory; however, their study is deductive from the Theory of Change and they have not reported nor demonstrated that this study is seeking to develop a theory. Furthermore, information regarding the methods to support a grounded theory methodology is missing. I would recommend that the authors provide further information regarding data collection (who conducted the interviews and focus groups? How were these conducted?) and data analysis (how was the analysis iterative? How did the authors analyze within and across themes or categories? Did the authors employ axial coding?). I would also suggest that the authors explain how and why the interviews were validated and reviewed for quality assurance. Lastly, I would suggest that the authors provide a qualitative research question and sub-questions (if relevant). Please find further minor recommendations below.

Abstract: Under methods, I would suggest rewording “the value of girls” for the violence prevention outcome.

Background: Thank you for expanding on the importance of adolescent development. I would suggest reframing that this developmental stage is inherently vulnerable to instead underscore the reasons for vulnerability that you’ve identified in the second sentence of your second paragraph (“lack of economic security, unequal gender norms, pressure from peers to engage in sexual activity, pressure from families to achieve economic security through early marriage, and not living with one’s parents”). I would also recommend rewording or contextualizing the “cascading ramifications of pregnancy” in this section.

Methods: As indicated above, I strongly suggest that the authors revise their qualitative methodology section. In addition, I would recommend that the authors provide an account of the theoretical underpinnings of the Theory of Change that is guiding this study. I also wonder if the authors would not consider reframing the qualitative component of this study as a qualitative evaluation of the trial – this reframing would seem to be appropriate given the aims of the qualitative exploration that they have identified.

Ethical considerations: Can the authors please provide an explanation for why parental assent was required for participants between 12-18 years of age?

Results: I would suggest that the authors begin the results section with an account of their participants’ demographic information and sample sizes. Although the participants provided further context for their qualitative findings, there are few quotes included to support the analysis; this may be due to limited space. I would therefore recommend that the authors include a table of qualitative findings, including themes and participant quotes. In Table 2, results for a measured cognitive score are presented – can the authors please speak to the purpose of this measure for the study. I also wondered why participants in the V-only arm did not receive transfers and ask that the authors briefly provide an explanation for this. Lastly, I would strongly recommend that the authors revise the language used to explain that girls learned to “protect themselves from boys and against violence”, as this wording currently places the responsibility of experiences of violence on girls.

7. PLOS authors have the option to publish the peer review history of their article (what does this mean?). If published, this will include your full peer review and any attached files.

Reviewer #1: No

Reviewer #2: No

---

## [Author Response · Author response to Decision Letter 1]

15 Dec 2021

Author response to second round of reviews for PONE-D-21-06054R1

We thank the reviewers for the additional valuable comments and have redrafted the paper to address them. 

Response: We have reviewed all the references and made some amendments to ensure that they are accurately presented. Please let us know if there are any specific reference you feel need further attention.

Due to the changes we have made as a result of suggestions made by the reviewers, we have retracted one reference: Strauss, A. and J. Corbin. 1998. Basics of Qualitative Research: Techniques and Procedures for Developing Grounded Theory. Thousand Oaks, CA: Sage Publications.

Reviewer #1: Although there is some improvement in manuscript after revision, still there are many confusing statements and as said earlier the ‘presentation’ is not ‘precise’, to-the-point. Please check the English language [professional intervention may be needed, I guess]. I have no specific recommendation(s) and think that, ‘let the respected editor decide the future course’.

Response: After incorporating material and reworking to address Reviewer #2’s comments we 1) reviewed and edited the entire manuscript ourselves and then 2) had it reviewed and edited by the team in the Population Council Inc Knowledge Communications office. 

Reviewer #2: Thank you to the authors for their in-depth responses to reviewer comments and for revising this manuscript accordingly. My remaining major recommendations are as follows: (1) the authors speak to transactional sex in the background and prostitution in their findings. As it is written, the authors seem to be overlooking that children and youth cannot consent to sex work. Instead of reporting that girls in their sample engaged in prostitution, I would recommend writing that they have experience child sexual exploitation. If the context and laws in Kenya suggest otherwise, I strongly recommend that the authors highlight this;...

Response: Thank you for this suggestion. We agree with the reviewer and in line with their suggestion have replaced ‘transactional sex’ in the background and ‘prostitution’ in the findings section with child sexual exploitation. 

...and (2) the section describing qualitative methodology and methods is still insufficient. The authors reported using grounded theory; however, their study is deductive from the Theory of Change and they have not reported nor demonstrated that this study is seeking to develop a theory. Furthermore, information regarding the methods to support a grounded theory methodology is missing. 

Response: Thank you for this comment. We have re-framed the description of the qualitative component as a qualitative evaluation of the trial and clarified the description of the approach to the analysis that was undertaken. 

I would recommend that the authors provide further information regarding data collection (who conducted the interviews and focus groups? How were these conducted?) and data analysis (how was the analysis iterative? How did the authors analyze within and across themes or categories? Did the authors employ axial coding?). I would also suggest that the authors explain how and why the interviews were validated and reviewed for quality assurance. Lastly, I would suggest that the authors provide a qualitative research question and sub-questions (if relevant). Please find further minor recommendations below.

Response: We have incorporated the additional details regarding the qualitative data collection in the qualitative methodology section. All interviews were conducted by trained interviewers and moderators. The analyses consisted firstly of developing a start-list of codes that were derived from the program’s Theory of Change as well as from the interview guides. Further themes were added as they emerged from reviewing the data. To ensure quality assurance, all transcripts were double coded by two qualified analysts. All double-coded transcripts underwent testing for intercoder agreement and where the Krippendorff’s c-α-binary coefficient was below 0.70, a side-by-side comparison, clarification and reconciliation was carried out on the specific coded transcripts.

Abstract: Under methods, I would suggest rewording “the value of girls” for the violence prevention outcome.

Response: Thank you for this suggestion. We have now rephrased “the value of girls” to “unequal gender norms and their consequences” which we feel is a more accurate reflection of what was discussed in the community dialogues. 

Background: Thank you for expanding on the importance of adolescent development. I would suggest reframing that this developmental stage is inherently vulnerable to instead underscore the reasons for vulnerability that you’ve identified in the second sentence of your second paragraph (“lack of economic security, unequal gender norms, pressure from peers to engage in sexual activity, pressure from families to achieve economic security through early marriage, and not living with one’s parents”). I would also recommend rewording or contextualizing the “cascading ramifications of pregnancy” in this section.

Response: We have modified the first paragraph so that it more clearly describes how developmental changes that occur during adolescence, in combination with certain negative external factors, increase the risk of early pregnancy. We have in addition changed the phrase “cascading ramifications of pregnancy” to “negative potential consequences of pregnancies.”

Methods: As indicated above, I strongly suggest that the authors revise their qualitative methodology section. In addition, I would recommend that the authors provide an account of the theoretical underpinnings of the Theory of Change that is guiding this study. 

Response: We have made revisions to the qualitative methodology section. In addition, we now reference the theories used to develop the original AGI-K theory of change and provide relevant citations.

I also wonder if the authors would not consider reframing the qualitative component of this study as a qualitative evaluation of the trial – this reframing would seem to be appropriate given the aims of the qualitative exploration that they have identified. 

Response: Thank you for this suggestion, we agree this is a better characterization of the work. We have made the changes to the ‘Randomization and Data Collection’ and ‘Qualitative Methodology’ sections to indicate that the qualitative component of the study was indeed a qualitative evaluation of the program. 

Ethical considerations: Can the authors please provide an explanation for why parental assent was required for participants between 12-18 years of age?

Response: Our description was unclear, and we have edited it. Written informed consent was required from all girls 18 years old or older. For girls under 18 years old, written informed consent was required from a parent or guardian, and oral assent from the girl herself. 

Results: I would suggest that the authors begin the results section with an account of their participants’ demographic information and sample sizes. Although the participants provided further context for their qualitative findings, there are few quotes included to support the analysis; this may be due to limited space. I would therefore recommend that the authors include a table of qualitative findings, including themes and participant quotes. 

Response: We have added in a section on participants’ demographic information and sample sizes at the beginning of the qualitative results section. Moreover, we include a table to display the socio-demographic characteristics of the participants that were interviewed for the qualitative evaluation (Table 5). Additional quotes have been incorporated into the qualitative findings section. 

In Table 2, results for a measured cognitive score are presented – can the authors please speak to the purpose of this measure for the study.

Response: Because an important secondary outcome in the study was education, we measured cognitive scores to enable careful assessment of baseline balance on an important indicator associated with schooling advancement. The cognitive score measure is used as a control in the extended controls models to increase precision of the estimates. The theory of change did not posit that the intervention would strongly influence cognitive scores, so it was measured only at baseline. 

I also wondered why participants in the V-only arm did not receive transfers and ask that the authors briefly provide an explanation for this. 

Response: An objective of the study was to provide rigorous evidence on the causal impacts of providing the three different packages of interventions (all with the education component) against a comparison group. To do so, we used a randomized trial research design in which girls were randomly allocated to four different study arms, including a study arm without the education, health or wealth interventions to be used as the experimental comparison in the quantitative analyses. 

Lastly, I would strongly recommend that the authors revise the language used to explain that girls learned to “protect themselves from boys and against violence”, as this wording currently places the responsibility of experiences of violence on girls.

Response: Thank you for this suggestion. We have amended the phrase “protect themselves from boys and against violence” to “recognize and protect themselves from sexual and gender-based violence”

---

## [Editor Report · Decision Letter 2]

7 Jan 2022

Impacts of multisectoral cash plus programs after four years in an urban informal settlement: Adolescent Girls Initiative-Kenya (AGI-K) randomized trial

PONE-D-21-06054R2

Dear Dr. Kangwana,

We’re pleased to inform you that your manuscript has been judged scientifically suitable for publication and will be formally accepted for publication once it meets all outstanding technical requirements.

Kind regards,

Catherine E Oldenburg

Academic Editor

PLOS ONE
---

## [Editor Report · Acceptance letter]

21 Jan 2022

PONE-D-21-06054R2 

Impacts of multisectoral cash plus programs after four years in an urban informal settlement: Adolescent Girls Initiative-Kenya (AGI-K) randomized trial 

Dear Dr. Kangwana:

I'm pleased to inform you that your manuscript has been deemed suitable for publication in PLOS ONE. Congratulations! Your manuscript is now with our production department. 

Kind regards, 

on behalf of

Dr. Catherine E Oldenburg 

Academic Editor

PLOS ONE